# The landscape of transcriptional and translational changes over 22 years of bacterial adaptation

**John S Favate[1]\*, Shun Liang[1], Alexander L Cope[1,2], Srujana S Yadavalli[1,3], Premal Shah[1,4]\***

[1]Department of Genetics, Rutgers University, Piscataway, United States; [2]Robert Wood Johnson Medical School, Rutgers University, New Brunswick, United States; [3]Waksman Institute, Rutgers University, Piscataway, United States; [4]Human Genetics Institute of New Jersey, Rutgers University, Piscataway, United States

**Abstract** Organisms can adapt to an environment by taking multiple mutational paths. This redundancy at the genetic level, where many mutations have similar phenotypic and fitness effects, can make untangling the molecular mechanisms of complex adaptations difficult. Here, we use the *Escherichia coli* long-term evolution experiment (LTEE) as a model to address this challenge. To understand how different genomic changes could lead to parallel fitness gains, we characterize the landscape of transcriptional and translational changes across 12 replicate populations evolving in parallel for 50,000 generations. By quantifying absolute changes in mRNA abundances, we show that not only do all evolved lines have more mRNAs but that this increase in mRNA abundance scales with cell size. We also find that despite few shared mutations at the genetic level, clones from replicate populations in the LTEE are remarkably similar in their gene expression patterns at both the transcriptional and translational levels. Furthermore, we show that the majority of the expression changes are due to changes at the transcriptional level with very few translational changes. Finally, we show how mutations in transcriptional regulators lead to consistent and parallel changes in the expression levels of downstream genes. These results deepen our understanding of the molecular mechanisms underlying complex adaptations and provide insights into the repeatability of evolution.

**\*For correspondence:**
john.favate@rutgers.edu (JSF);
premal.shah@rutgers.edu (PS)

## Editor's evaluation

This paper comprehensively analyzes how gene expression has changed in eleven *E. coli* strains after 50,000 generations of laboratory evolution. It confirms that, overall, changes in RNA levels are more reproducible than the underlying genetic changes and begins to investigate how some of these changes lead to increased fitness in this environment. This dataset will be a valuable resource for testing theories about how genotypic and phenotypic evolution are coupled and for understanding how bacterial gene regulatory networks evolve during adaptation.

## Introduction

A key challenge in biology is understanding the relationships between genotype, phenotype, and evolutionary fitness. Comparative genomic approaches and large-scale mutation experiments have allowed us to map genetic changes to phenotypic changes underlying adaptation. For example, mutations that increase the affinity of hemoglobin for oxygen are adaptive in high-altitude dwelling deer mice (*Natarajan et al., 2013*), and mutations to the influenza haemagglutinin and neuraminidase proteins increase viral fitness (*Gong et al., 2013*; *Lee et al., 2018*). Adaptive phenotypes can also

**eLife digest** The reason we look like our parents is because we inherit their genes. Genes carry the instructions for our cells to make messenger RNAs (mRNAs), which our cells then translate into proteins. Proteins, in turn, determine many of our features. This is true for all living organisms. Any changes – or mutations – in an organism's genes can lead to variations in its proteins, which can alter the organism's traits. This is the basis for evolution: mutations can lead to changes that allow an organism to better adapt to a new environment. This increases the organism's chances of survival and reproduction – its evolutionary 'fitness' – and makes it more likely that the mutation that generated the new trait in the first place will be passed on to the organism's descendants.

However, just because two organisms have evolved similar traits to adapt to similar environments, it does not mean that the genetic basis for the adaptation is the same. For example, many animals share similar coloring to warn off predators, but the way that coloring is coded genetically is completely different. In species that are related (which share many of the same genes), this type of evolution is called 'parallel evolution', and it can make it difficult for scientists to understand how an organism evolved and pinpoint exactly what mutations are linked to which features.

In 1988, scientists established the 'long-term evolution experiment' to tackle questions about how evolution works. The experiment, which has been running for over 30 years, consisted on tracking the evolution of 12 populations of *Escherichia coli* bacteria grown in separate flasks containing the same low-nutrient medium. The initial 12 populations were genetically identical, making this an ideal system to study parallel evolution, since all the populations had to evolve to adapt to the same environment, whilst isolated from each other. In previous experiments, scientists had already noted that while the different bacterial populations grew in similar ways, they had mostly different mutations.

To better understand parallel evolution, Favate et al. analyzed the synthesis rates of RNA and proteins in the *E. coli* populations used in the long-term evolution experiment. They found that 22 years after the start of the experiment, all 12 populations produced more RNA, grew faster and were bigger. Additionally, while the different populations had accumulated few shared mutations after 22 years, they all shared similar patterns of RNA levels and protein synthesis rates. Further probing revealed that parallel evolution may be linked to how genes are regulated: mutations in regulators of related groups of genes involved in the same processes inside the cell can amplify the degree of parallel changes in organisms. This means that mutations in these genes may lead to similar traits.

These findings provide insight into how parallel evolution arises in the long-term evolution experiment, and provides clues as to how the same traits can evolve several times.

result from changes in multiple genes, such as in yeast evolving under nutrient limitation (*Gresham et al., 2008*; *Lauer et al., 2018*; *Venkataram et al., 2016*), bacterial adaptation during infection (*Lieberman et al., 2011*) or to high temperature (*Tenaillon et al., 2012*), and in the evolution of smaller body sizes in Atlantic silversides under a size-selective fishing regime (*Therkildsen et al., 2019*). In many cases, similar adaptive phenotypes arise from different mutations to the same gene or regulatory region or from combinations of mutations to different genes and regulatory regions. This redundancy, where many genotypes produce similar phenotypes, makes it difficult to understand the molecular mechanisms behind adaptive phenotypes and is exacerbated by potential epistatic interactions among mutations. On the other hand, adaptive changes to expression have been shown to occur during the domestication of eggplants and tomatoes (*Koenig et al., 2013*; *Page et al., 2019*), and in hybridization events between two weeds (*Kryvokhyzha et al., 2019*). Although not direct observations of adaptive changes to gene expression, recent comparative analyses of across-species gene expression suggest that the expression levels of numerous genes are evolving under directional selection in vertebrates, fish, and butterflies (*Brawand et al., 2011*; *Catalán et al., 2019*; *Fukushima and Pollock, 2020*; *Gillard et al., 2021*).

Here, we use the long-term evolution experiment (LTEE) (*Lenski et al., 1991*) as a model to characterize the molecular changes underlying adaptation to a novel environment. In the LTEE, 12 replicate populations of *Escherichia coli* have been adapting in parallel to a carbon-limited medium since 1988, growing over 75,000 generations thus far. As is common in lab-based evolution experiments, the replicate populations display similar phenotypic changes (*Blount et al., 2018*). Examples include increases

in fitness (*Wiser et al., 2013*) and cell size (*Grant et al., 2021*; *Philippe et al., 2009*). In contrast, a significant amount of diversity exists at the genomic level across the replicates (*Tenaillon et al., 2016*), with some lines having orders of magnitude more mutations than others due to the development of mutator phenotypes (*Good et al., 2017*). While few mutations are shared at the nucleotide level, some genes are commonly mutated across evolved lines (*Maddamsetti et al., 2017*; *Woods et al., 2006*). Still, how most of the mutations affect fitness in the system is unknown.

Researchers have hypothesized that similar gene expression patterns might contribute to the parallel increases in fitness in the LTEE (*Fox and Lenski, 2015*). An earlier microarray-based study of transcriptional changes in LTEE showed parallel changes in mRNA abundances in clones from two evolved lines (Ara-1 and Ara+1) at 20,000 generations (*Cooper et al., 2003*). However, it remained unclear which mutations were responsible for these parallel changes and whether the remaining 10 lines also had similar expression profiles.

Moreover, protein-coding mRNAs must be translated to perform their function. The majority of cellular biomass and energy expenditure is devoted to translation (*Bernier et al., 2018*), and the role of hierarchical regulation of gene expression in evolutionary processes has been a subject of debate in recent years (*Albert et al., 2014*; *Artieri and Fraser, 2014*; *McManus et al., 2014*). However, we know little of changes in gene expression at the translational level in the LTEE.

Here, we use both RNA-seq and Ribo-seq (*Ingolia et al., 2009*) to profile the landscape of transcriptional and translational changes after 22 years (50,000 generations) of evolution in the LTEE to answer five fundamental questions: (i) do evolved lines show similar transcriptomic and translatomic changes after 50,000 generations despite acquiring mostly unique sets of mutations? (ii) how do changes in cell size affect changes in absolute expression levels? (iii) do changes in gene expression at the translational level buffer, augment, or match changes at the transcriptional level?, (iv) what classes of genes or pathways are altered in the evolved lines, and finally, (v) can we identify mutations responsible for parallel changes in gene expression across replicate populations?

## Results

We generated RNA-seq and Ribo-seq datasets for single clones grown in the exponential phase from each of the 12 evolved lines with sequenced genomes in *Tenaillon et al., 2016* (see Materials and methods section M1 for specific clone IDs) (*Figure 1A*). We aligned each clone's data to its unique genome and considered expression changes of 4131 transcripts from the ancestor. Due to concerns of contamination in our Ara+6 samples, we removed them from further analysis. We averaged between 151 and 1693 deduplicated reads per transcript across the 52 libraries (*Figure 1—figure supplement 1A*, *Supplementary file 1*), the distributions of read counts per transcript were similar across lines, replicates, and sequencing methods (*Figure 1—figure supplement 1B*), and correlations between biological replicates were high (Pearson correlation coefficient $R>0.93$, *Figure 1—figure supplement 1C*). We also verified the presence of three-nucleotide periodicity in our Ribo-seq datasets (*Figure 1—figure supplement 1D*). Previous studies have shown the existence of distinct ecotypes in the Ara-2 population (*Plucain et al., 2014*; *Rozen et al., 2009*). Based on an analysis of mutations, our Ara-2 clone comes from the L ecotype (see Appendix A1). Our Ara-3 clone can utilize citrate as a carbon source (Cit+). Finally, we note that both ancestral and evolved lines were grown in standard LTEE media supplemented with additional glucose to obtain enough starting material for paired RNA-seq and Ribo-seq samples. We discuss the potential impacts of this difference in the supplement (Appendix A2).

### Evolved lines show parallel transcriptomic changes
### Gene expression levels are similar across evolved lines

Across the six evolved lines with non-mutator phenotypes in LTEE, we observe a modest degree of parallelism in genetic changes. We find that 22 genes share mutations in two or more evolved lines (*Tenaillon et al., 2016*). However, it remains unclear whether these parallel genetic changes are sufficient to explain the high degree of parallelism in fitness gains over 50,000 generations. We hypothesize that the evolved lines demonstrate a higher degree of parallel transcriptomic changes despite having unique genomes. To test this hypothesis, we compared the ancestors' and evolved lines' mRNA abundances (measured in transcripts per million [TPM]). We find that the expression levels of most

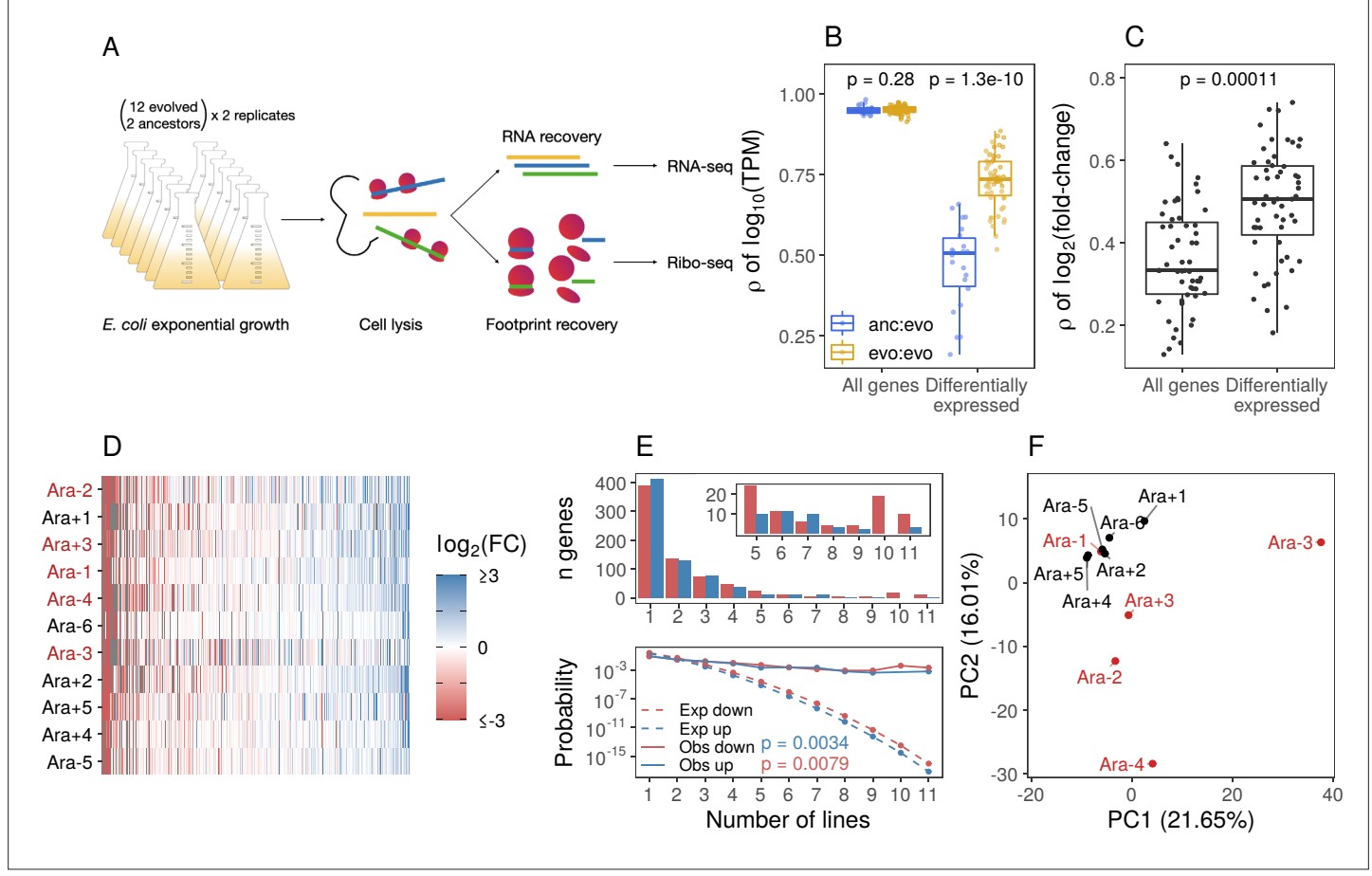

**Figure 1.** Parallel changes in mRNA abundances. (**A**) Schematic diagram of the experimental setup. (**B**) Pairwise Pearson correlations based on $log_{10}(TPM)$ (where transcripts per million [TPM] is the mean from replicates) separated by comparisons between evolved lines or from ancestors to evolved lines. p-Values indicate the results of a Kolmogorov-Smirnov (KS) test. For differentially expressed genes (DESeq2 q ≤ 0.01), evolved line were compared using the union of the significant genes from each line. When comparisons were between an evolved line and an ancestor, the significant genes from that evolved line were used. (**C**) Pairwise Spearman's correlations based on fold-changes from all genes, and the union of the significant genes between two evolved lines (differentially expressed). (**D**) Fold-changes of differentially expressed genes that were significantly altered in at least one line. Genes are ordered left to right in increasing mean fold-change across all evolved lines. Genes containing deletions are not assigned a fold-change and are represented as gray spaces. Lines with a mutator phenotype are in red. (**E**) The upper panel shows the number of genes (y-axis) that were both statistically significant and had a fold-change in the same direction in a particular number of lines (x-axis). The bottom panel shows the expected (dashed) and observed (solid) probability of observing a particular result. p-Values are the result of a KS test between the observed and expected distributions. (**F**) Principal component analysis (PCA) based on all fold-changes. In this case, genes with some form of deletion (complete or indel) are assigned a fold-change of −10 to indicate severe downregulation because they are either completely absent from the genome or not expected to produce functional proteins.

The online version of this article includes the following figure supplement(s) for figure 1:

**Figure supplement 1.** Sequencing data statistics.

**Figure supplement 2.** Magnitude and variation in mRNA fold-changes across evolved lines.

**Figure supplement 3.** Comparison of expression changes between this study and *Cooper et al., 2003*.

genes remain unchanged, leading to high correlations between ancestral and evolved strains (Spearman's correlation coefficient *r*>0.95, *Figure 1B*). Moreover, pairwise correlations between evolved strains were only marginally higher than the correlations between evolved strains and the ancestors. However, these increases were not statistically significant (KS test, p-value = 0.28, *Figure 1B*). This suggests that transcriptomic changes are likely restricted to a small portion of the genome.

To more formally test the hypothesis that evolved lines show parallel changes in the transcriptome, we used DESeq2 (*Love et al., 2014*) to identify differentially expressed genes (DEGs) and quantify expression changes between each evolved line and the ancestor (for full results, see *Supplementary*

*file 2*). A gene was considered differentially expressed between the evolved line and the ancestor if it reached a statistical threshold of q-value ≤0.01. We find that most fold-changes were small (*Figure 1—figure supplement 2A*) and consistent with our expectations; only a small proportion of the transcriptome was significantly altered (*Figure 1—figure supplement 2B*). On average, ~270 genes (out of 4131) were differentially expressed in an evolved line across all 11 pairwise comparisons between each evolved line and the ancestor. In total, 2986 genes were differentially expressed, but this consisted of only 1273 unique genes, indicating that many DEGs are shared across evolved lines. The expression levels of these 1273 DEGs were more similar between evolved lines than between an evolved line and its ancestor (*Figure 1B*). Correlations based on fold-changes for DEGs were higher than those based on all genes (*Figure 1C*). Fold-changes for the set of 1273 DEGs were generally in the same direction regardless of their statistical significance (*Figure 1D*). Taken together, this is suggestive of parallelism in the evolution of gene expression across the evolved lines.

## Quantifying the degree of parallelism of DEGs

To test if the number of observed parallel changes in gene expression across evolved lines differs from the number of parallel changes expected by random chance, we estimated the probability distribution representing the expected number of DEGs altered in the same direction given different proportions of up- and downregulated genes in each line. This null distribution is well approximated by the distribution of the *sum of independent non-identical binomial random variables* (SINIB), which we estimated using the R package sinib (*Liu and Quertermous, 2018*) by parameterizing the function with the number of up- and downregulated DEGs from each line (*Figure 1—figure supplement 2C*). We find that the number of genes with expression changes in the same direction is significantly higher than expected by chance (KS test, p-value ~ 0.01, *Figure 1E* – bottom panel). For example, if DEGs were randomly distributed across all lines, we would expect three genes to share expression changes in five or more lines. Instead, 117 genes are differentially expressed in the same direction in at least five lines.

## Magnitude and direction of expression changes

Given the high correlations between expression levels of DEGs between evolved lines, it stands to reason that the correlation between fold-changes of DEGs genes will be higher than the correlation between fold-changes across all genes. Consistent with these expectations, we find that pairwise correlations between evolved lines of fold-changes in DEGs were higher than the fold-changes of all genes (*Figure 1C*). While the number of DEGs varies widely across lines (*Figure 1—figure supplement 2B*), 7 out of 11 evolved lines have more significantly downregulated DEGs than upregulated (*Figure 1—figure supplement 2D*, binomial test, p-value <0.05). Furthermore, the magnitude of fold-changes of downregulated DEGs was significantly higher than fold-changes of upregulated DEGs in all 11 evolved lines (*Figure 1—figure supplement 2D*, KS test, p-value <0.0001).

## Variation in expression changes across evolved lines

So far, we have considered the degree of parallelism in expression level changes across the evolved lines. However, the evolved lines differ not only in terms of their underlying mutations (*Tenaillon et al., 2016*) but also vary substantially at the phenotypic level. For instance, half of the evolved lines have developed a mutator phenotype, causing them to accumulate orders of magnitude more mutations than the non-mutator lines. Unlike the other 11 evolved lines, Ara-3 can utilize citrate as a carbon source (*Blount et al., 2012*), and Ara-2 has developed distinct, coexisting ecotypes (*Rozen et al., 2009*). We wanted to characterize how phenotypic variation across evolved lines might correlate with variation in expression levels. Principal component analysis (PCA) based on all fold-changes mainly separates Ara-3 from the rest of the lines, whereas PC2 appears to separate at least some of the mutators from the non-mutators (*Figure 1F*). Variation in PC1 and PC2 seems primarily driven by deletions (*Figure 1—figure supplement 2E*), coded as downregulated genes (log2 fold-change = −10) in this analysis. The magnitude of encoded fold-changes of the deleted genes did not affect the groupings of the PCA between log2(fold-change) −1 and −10. Given the unique circumstances in Ara-3 and Ara-2, it is not surprising that these lines group separately from the others in the PCA.

## Evolved lines are larger in cell size and carry more mRNAs

In the previous section, we discussed how changes in relative gene expression patterns across the evolved lines are similar. However, all evolved lines are significantly larger than their ancestors (*Grant et al., 2021*; *Lenski and Mongold, 2000*; *Mongold and Lenski, 1996*). Typically, bacterial cell volume depends on nutrient availability and growth rate (*Chien et al., 2012*; *Schaechter et al., 1958*; *Taheri-Araghi et al., 2015*) and the increase in cell volume in evolved lines appears to be under selection rather than solely due to increases in growth rate (*Mongold and Lenski, 1996*; *Philippe et al., 2009*). As a result of these larger sizes, the cells in evolved lines have higher biomass and proportionally higher nucleic acid levels than the ancestors (*Turner et al., 2017*). Therefore, it is reasonable to expect that absolute abundances of mRNA molecules per cell should also increase with cell volume to maintain concentrations and reaction rates (*Padovan-Merhar et al., 2015*). To get a complete picture of transcriptional changes, we also quantified absolute changes in mRNA abundances.

We used phase-contrast microscopy to measure cell shape and estimate cell volume to confirm that our clones from evolved lines were larger than their ancestors (see Appendix A3). Consistent with earlier studies, we find that each evolved line is larger in volume compared to its ancestors (*Figure 2A*, *Supplementary file 3*). Our volume estimates are also consistent with measurements obtained using a Coulter counter from a recent study (*Grant et al., 2021*; *Figure 2—figure supplement 1A*, Pearson correlation coefficient $R$=0.87). Next, we estimated the absolute abundances of transcripts per CFU by comparison to known standards in our sequencing libraries. Specifically, we added the ERCC spike-in controls (*Baker et al., 2005*; *External RNA Controls Consortium, 2005*) to our sequencing libraries and used a linear model to relate the number of molecules of a spike-in RNA added to its TPM in each sample. We find a linear relationship between molecules added and estimated TPM across all samples and replicates (*Figure 2B*, *Figure 2—figure supplement 2A*, *Supplementary file 5*). Finally, we measured the number of cells used in the generation of each sequencing library by counting colony-forming units (CFUs) from each culture and accounting for sampling at each step of the library preparation (*Supplementary file 4*). Note that due to various factors, our estimates of CFU are likely underestimates (see Appendix A3, *Figure 2—figure supplement 1C*). Nonetheless, our gene-specific estimates of absolute abundances per CFU are highly similar across biological replicates ($R$>0.93). Together, this allows us to measure absolute RNA abundance per CFU.

We find that most genes have increased mRNA abundance per CFU compared to the ancestor (*Figure 2C*, *Figure 2—figure supplement 2B*, *Supplementary file 6*) and that these differences were significantly larger than the differences between biological replicates (*Figure 2D*). Furthermore, the increases in total mRNA abundance scale with cellular volume, with larger evolved lines having more molecules per typical cell volume (*Figure 2E*). This suggests that the evolved lines have more mRNA per cell than the ancestors. Such an increase may be needed to maintain reaction rates in the face of increasing cell volumes. Another hypothesis is that stockpiling resources like mRNA and ribosomes might allow evolved lines to reduce the time spent in the lag phase after transfer to fresh medium. Indeed, reduced lag times occur in the LTEE (*Vasi et al., 1994*), and simulations suggest that bacteria can evolve to 'anticipate' the regular transfer to fresh medium in a serial transfer regime (*van Dijk et al., 2019*).

## Transcriptional changes drive translational changes

While mRNA abundances are an important molecular phenotype potentially linking genomic changes to adaptations, changes in mRNA abundances can themselves be buffered or augmented at other downstream regulatory processes such as translation (*Albert et al., 2014*; *Artieri and Fraser, 2014*; *McManus et al., 2014*). Translational regulation affects the rate at which an mRNA produces its protein product, and mRNAs vary widely in their translation efficiencies in both eukaryotes and prokaryotes (*Ingolia et al., 2009*; *Li et al., 2014*; *Picard et al., 2012*). However, the role of changes in translational regulation during adaptation and speciation remains poorly understood and, at least in yeast, is heavily debated (*Albert et al., 2014*; *Artieri and Fraser, 2014*; *McManus et al., 2014*). Moreover, because translation occupies the majority of cellular resources (*Bernier et al., 2018*), it may be a prime target for evolution in the LTEE. To study translational changes in LTEE, we performed Ribo-seq in the evolved lines and their ancestors (*Figure 1A*).

We find that changes in ribosome densities are highly correlated with changes in mRNA abundances (*Figure 3A*, *Figure 3—figure supplement 1A*). This is somewhat surprising because changes

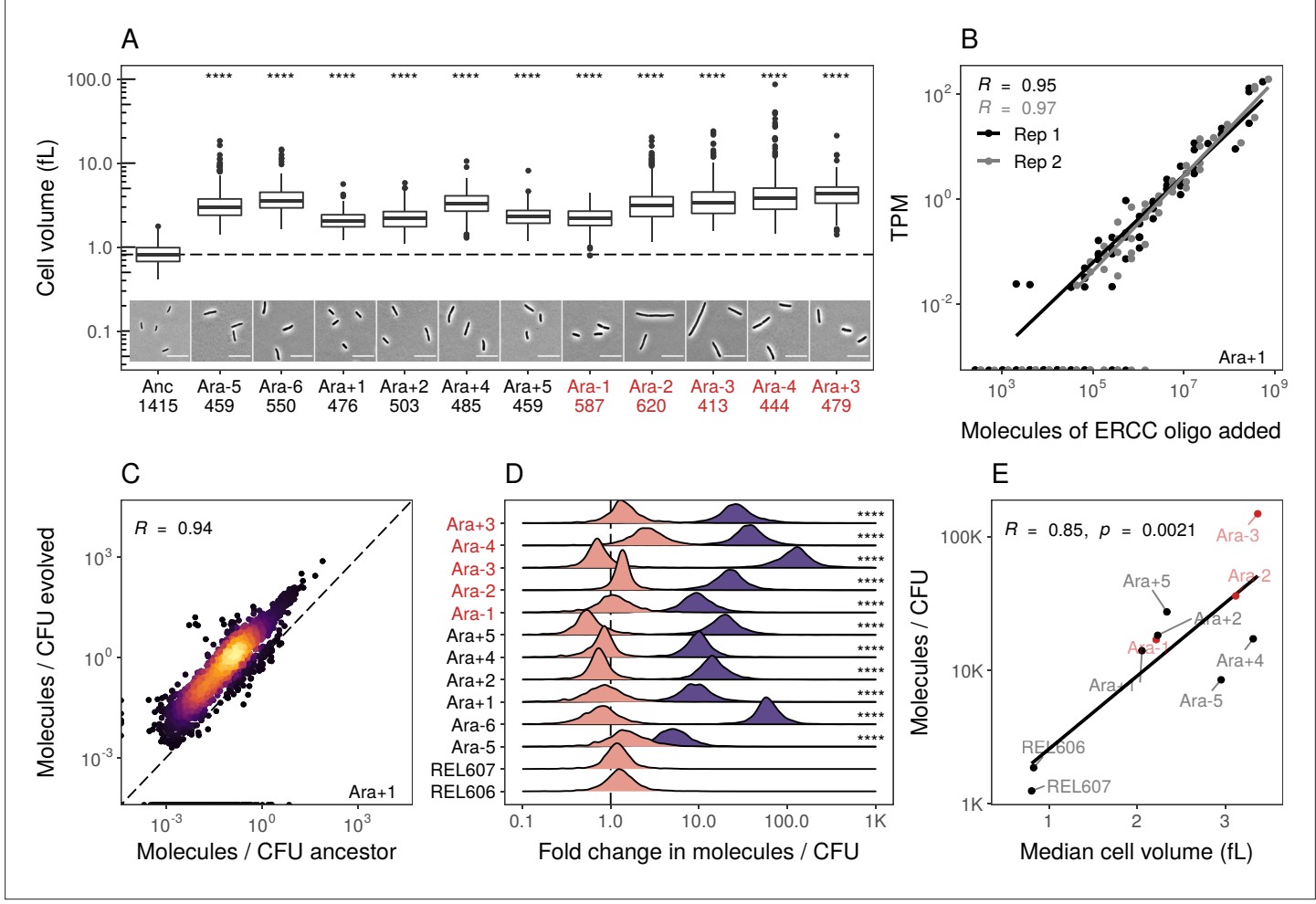

**Figure 2.** Evolved lines are larger in cell size and carry more mRNAs. (**A**) All evolved lines are larger than the ancestral strain. Distributions of cellular volume as determined by phase-contrast microscopy and assuming sphero-cylindrical shape of *Escherichia coli* along with representative images for each line. Numbers underneath a line's name indicate the total number of cells imaged (scale bar is 10 μm). The dashed line indicates the ancestral median, p-values indicate the results of a t-test when each line is compared to the ancestor, **** p ≤ 0.0001. Lines listed in red have mutator phenotypes. (**B**) Abundances of spike-in RNA control oligos are correlated with their estimates in sequencing data. Linear models relating the number of molecules of each ERCC control sequence added to their RNA-seq TPM (transcripts per million) in Ara+1 RNA-seq sample (see *Figure 2—figure supplement 2* for data for all lines). (**C**) Most genes have a higher absolute expression in evolved lines. Changes in the absolute number of mRNA molecules per CFU (colony-forming unit) in the 50,000th generation of Ara+1 relative to the ancestor. The values plotted are the averages between two replicates of the evolved lines and both replicates from two ancestors (REL606 and REL607; see *Figure 2—figure supplement 2* for all lines). (**D**) Absolute changes in mRNA abundances of genes in evolved lines are significantly larger than the variation between biological replicates (KS test, p<0.0001 in all cases). Pink distributions indicate gene-specific fold-changes between biological replicates for each line (centered around 1). Purple distributions show the absolute fold-changes in molecules of RNA per CFU from the ancestor to each evolved line. Fold-changes are calculated in the same manner as in C. (**E**) Larger evolved lines have more mRNA per CFU. Relationship between the median cellular volume for each line and the total number of RNA molecules per CFU. Total molecules of RNA are calculated as the sum of the average number of molecules for each gene between replicates.

The online version of this article includes the following figure supplement(s) for figure 2:

**Figure supplement 1.** Relationship between cellular features and cell volume.

**Figure supplement 2.** Absolute changes in mRNA abundances per CFU across all evolved lines.

in environmental conditions and small genetic perturbations usually result in large changes at the translational level (*Gerashchenko et al., 2012*; *Rubio et al., 2021*; *Woolstenhulme et al., 2015*). Despite the high correlation between mRNA and ribosome footprint fold-changes at the genomic level, individual genes might have altered ribosome densities. We used Riborex to quantify changes in ribosome densities (*Li et al., 2017*). Riborex quantifies changes in footprint densities while accounting for any changes in mRNA abundances. We considered a gene significantly altered if it reached a

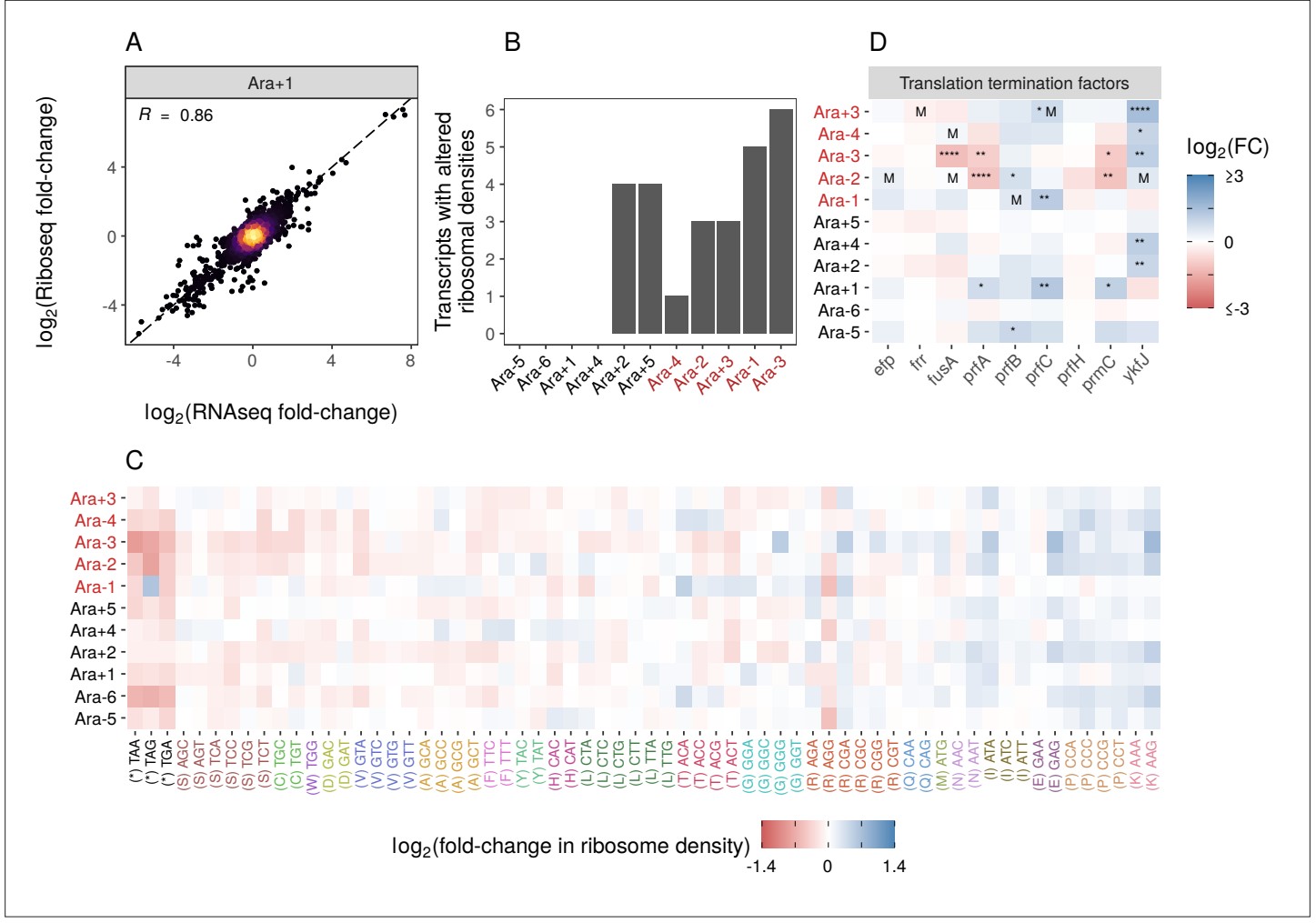

**Figure 3.** Changes in gene expression at the translational level. (**A**) Translational changes are correlated with transcriptional changes. The relationship between RNA-seq and Ribo-seq fold-changes in Ara+1 (see *Figure 3—figure supplement 1A* for all evolved lines). (**B**) The distribution of genes with significantly altered ribosome densities ($q \le 0.01$) estimated using Riborex ($q \le 0.01$). (**C**) Evolved lines have faster translation termination. Stop codons had lowered ribosome density compared to all sense codons. Changes in codon-specific ribosome densities in each of the evolved lines relative to the ancestor. Codons are colored according to the amino acid they code for. Amino acids are ordered left to right in order of mean fold-change across the lines. (**D**) Fold-changes in mRNA abundances of translation termination factors and related genes *ykfJ*, *prfH*, *prfA*, *prmC*, *prfB*, *fusA*, *efp*, *prfC*. RNA-seq fold-changes for termination factors, asterisks indicate DESeq2 q-values (blank: p > 0.05, *: $p \le 0.05$, **: $p \le 0.01$, ***: $p \le 0.001$ ****: $p \le 0.0001$ and an 'M' indicates an SNP in that gene).

The online version of this article includes the following figure supplement(s) for figure 3:

**Figure supplement 1.** Relationship between RNAseq and riboseq fold-changes for all evolved lines.

q-value ≤0.01. Only a handful of genes have altered ribosome densities, and none are shared between three or more lines (*Figure 3B*, *Supplementary file 7*). This suggests that over the course of the LTEE, most changes happen at the transcriptional level with insufficient evidence for significant changes at the translational level. We note that earlier studies have indicated that Riborex has limited power to detect small to moderate shifts in ribosome densities based on simulated data (*Li et al., 2017*). Although comparing these simulations to our data is difficult, it is possible that we are failing to detect some of these smaller shifts in gene-specific ribosome densities. Regardless, our results indicate a greater role for changes in factors regulating mRNA abundances than factors regulating mRNA translation.

While Riborex can find gene-specific changes in ribosome densities, Ribo-seq data can also provide codon level resolution, allowing us to perform a detailed analysis of the translation of specific codons or amino acids. We calculated genome-wide average codon-specific ribosome densities

(see Codon-specific positioning of Ribo-seq data in Materials and methods, *Supplementary file 8*) in each of our ancestral and evolved lines and observed a high correlation between replicates (Pearson correlation coefficient $R>0.98$). When comparing codon densities from each evolved line to the ancestor (*Figure 3C*), we find that densities at stop codons were lower in evolved lines than in the ancestor, indicating potentially faster translation termination. Importantly, ribosome densities estimated from the same evolved line are not truly independent, violating the assumption of independence for common statistical tests. We used a linear mixed model to account for possible evolved line-specific effects. The linear mixed model fit indicates an overall decrease in the ribosome density at stop codons relative to the sense codons, with a mean change in ribosome density (i.e., mean log2 fold-changes between evolved and ancestral lines) of –0.32 and 0.005, respectively. Note that these values represent the population-level fixed effect slope ($\beta_1 = -0.325$, p<0.05) and population-level fixed effect intercept ($\beta_0 = 0.005$, p=0.4423), respectively. The population-level fixed intercept ($\beta_0 = 0.005$, p=0.4423) indicates the sense codons, on average, experienced no change in ribosome densities between the evolved and ancestral lines (i.e., the mean log fold-changes of ribosome densities was 0). In contrast, the population-level fixed slope ($\beta_1 = -0.325$, p<0.05) indicates that stop codons, on average, experienced a decrease in ribosome density between the evolved and ancestral lines (i.e., the mean log fold-change of stop codon ribosome densities was –0.325 units lower than the mean log fold-change of sense codon). Accounting for line-specific effects, the stop codon effect sizes for each evolved line range from –0.088 to –0.657 log fold-change units (relative to sense codons), indicating that stop codons in all evolved lines have a decreased ribosome density compared to the ancestor. This suggests that the translation termination rate increased across all evolved lines (relative to the ancestral line), but this increase was greater in some evolved lines than others. For Ara-1, the TAG codon shows increased density, unlike other lines. This leads to a near-zero random effect size for this line.

Translation termination is one of the rate-limiting steps in translation and is typically much slower than codon elongation rates. Therefore, faster termination might increase the ribosome recycling rates and eventually allow faster translation initiation and protein production (*Andersson and Kurland, 1990*; *Plotkin and Kudla, 2011*; *Shah et al., 2013*). We wondered if faster termination was due to changes in the expression of translation termination factors. While some termination factors show increased expression in some lines, no single gene shows a consistent pattern across all lines (*Figure 3D*). Notably, while faster translation termination may increase ribosome recycling and enable faster growth, it may come at the expense of altering a key regulatory mechanism in translational control. As a result, it remains unclear if these regulatory changes can evolve in more complex environments.

## Functional characterization of DEGs

Thus far, we have only considered the magnitude and source of parallelism in expression changes. In this section, we attempt to functionally characterize the altered genes, identify mutations that might be driving some of these expression changes, and determine how much higher-order entities such as metabolic pathways are altered across the evolved lines. To identify altered functional categories and pathways, we use function and pathway analysis tools such as GO (*Ashburner et al., 2000*), KEGG (*Kanehisa and Goto, 2000*), and the BioCyc database pathway perturbation score (PPS, higher numbers indicate stronger alterations to a pathway) (*Karp et al., 2019*) to assess these features (see Functional analysis in Materials and methods). Because our data suggest that changes in mRNA abundances are the driving force of change in the system, we present results from our RNA-seq data but note that similar results are obtained when using the Ribo-seq data as well (*Figure 4—figure supplements 1 and 2*). For this section, we treat genes that experienced some form of deletion (complete or containing indels) as downregulated (log2 fold-change = –10) because they no longer produce functional proteins.

Many functional categories were altered across the lines in the KEGG analysis (*Supplementary file 9*). Consistent with earlier microarray experiments (*Cooper et al., 2003*), we find that the flagellar assembly genes are significantly downregulated in 10 out of 11 evolved lines (*Figure 4A*). Consistent with increased growth rates, we also find that many categories related to biosynthetic and metabolic processes involving sugars or amino acids are upregulated. The biosynthesis of nucleotide sugars appears downregulated mainly due to the deletion of many of the genes involved in creating sugars

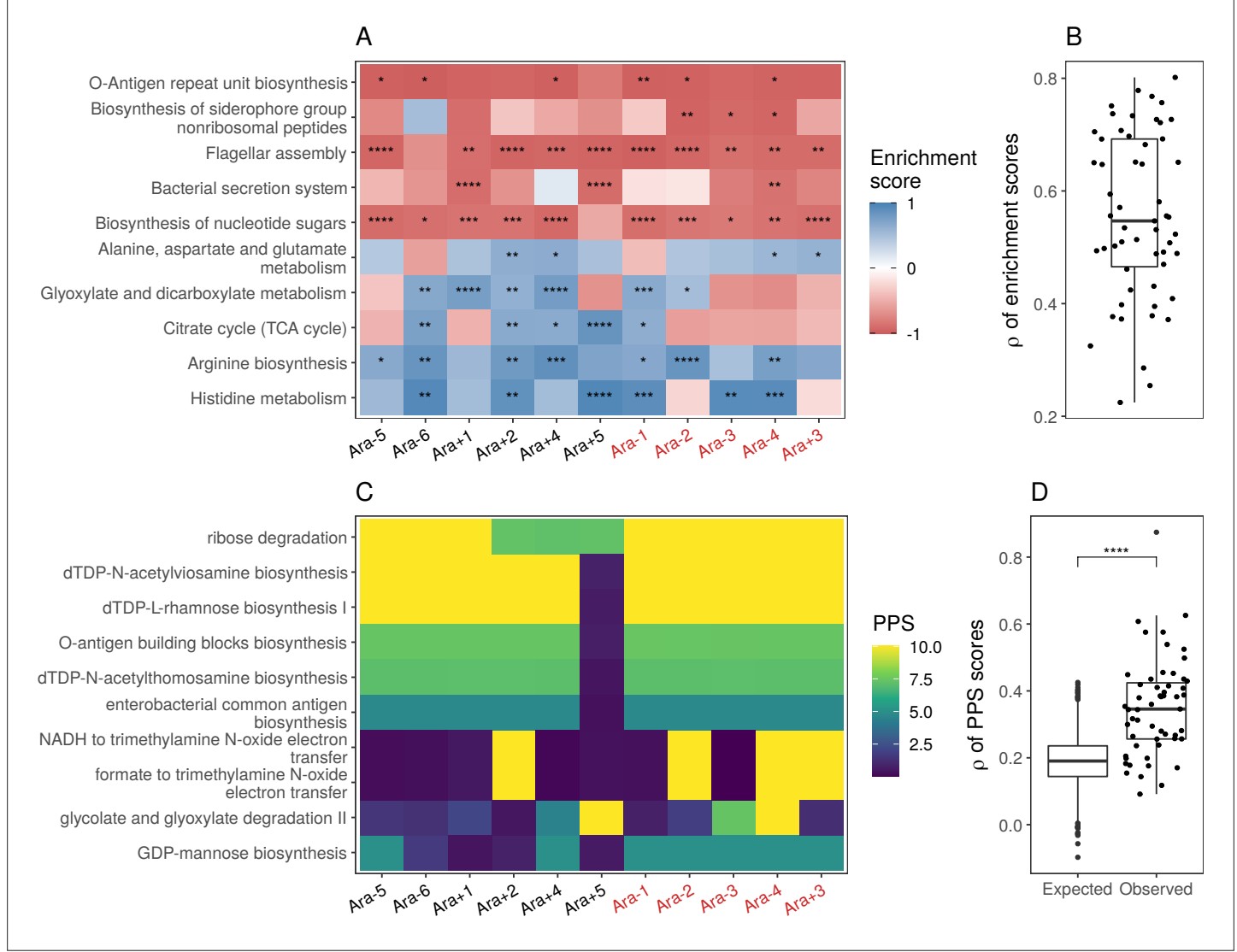

**Figure 4.** Parallel changes in biological processes and pathways. (**A**) Parallel changes in biological processes and pathways. The top 10 KEGG pathways that were significantly altered ($FDR \leq 0.05$) based on RNA-seq data. Enrichment score represents the degree to which a pathway was up- (positive) or downregulated (negative). Functional categories are ordered by increasing mean enrichment score across the lines. Enrichment score represents the degree to which a pathway was up- (positive) or downregulated (negative). (**B**) Distribution of pairwise Spearman's correlations of enrichment scores of all significantly altered functional categories ($FDR \leq 0.05$). (**C**) The top 10 pathways with the highest mean Pathway perturbation scores (PPS) calculated from RNA-seq fold-changes. Higher PPS indicates larger degrees of alteration but does not indicate directionality. (**D**) Distribution of pairwise Spearman's correlations based on all PPS (observed) compared to 1000 sets of correlations generated from PPS calculated after randomization of fold-changes (expected). The p-value is the result of a Kolmogorov-Smirnov test (blank: p > 0.05, *: p ≤ 0.05, **: p ≤ 0.01, ***: p ≤ 0.001 ****: p ≤ 0.0001).

The online version of this article includes the following figure supplement(s) for figure 4:

**Figure supplement 1.** Parallel changes in biological processes and pathways based on Ribo-seq data.

**Figure supplement 2.** GO and other functional analyses of differentially expressed genes.

which eventually lead to O-antigen biosynthesis. Many of these sugars are involved in constructing the cell membrane or walls; this could be related to known changes in cell shape and size (*Grant et al., 2021*). Overall, we find that changes in functional categories were mostly similar across all evolved lines (*Figure 4B*).

While KEGG pathway analysis encompasses molecular interactions and reaction networks, we wondered which specific metabolic reactions were altered across all lines and which ones remained unchanged over 50,000 generations. Because *E. coli* REL606 is annotated in the Biocyc collection of

databases, we used their metabolic mapping tool to score pathway alterations with a pathway perturbation score (PPS) in each of the evolved lines (see Functional analysis in Materials and methods for a detailed explanation of the scoring). Similar to the KEGG pathway analysis, we find a high degree of parallelism, even at the level of specific metabolic reactions (*Figure 4C and D*, *Figure 4—figure supplement 2D*). Interestingly, four out of five most altered pathways are involved in lipopolysaccharides biosynthesis, a major component of Gram-negative bacteria's outer membrane. This suggests that the composition of the evolved lines' outer membrane has significantly changed in addition to changes in cell size and shape. Nonetheless, there is a core set of unaltered pathways, even in clones with a mutator phenotype. Pathways with low PPS, indicating low levels of alteration, included D-serine degradation (mean RNA-seq PPS = 0.13, $\sigma$ = 0.07), pseudouridine degradation (mean RNA-seq PPS = 0.12, $\sigma$ = 0.06), and others (see *Supplementary file 11* for complete PPS). These may represent pathways with activity levels that cannot be altered or whose alteration provides little to no fitness benefit.

## Mutations to transcriptional regulators explain many parallel expression changes

Given the high degree of parallelism in evolved lines at the gene expression level, we wondered whether some of these patterns could be explained by a parallel set of mutations at the genetic level. Because KEGG, PPS, and GO analyses all identified metabolism and catabolism of various sugars to be significantly altered, we looked at mutations to genes involved in these functional categories. Previous work has shown that depending on the generation sampled, evolved clones grow poorly (20,000th generation) or not at all (50,000th generation) on maltose (*Leiby and Marx, 2014*). Because maltose is absent from the growth media in the LTEE, maintenance of these transporters is likely unnecessary (*Pelosi et al., 2006*). Additionally, at 20,000 generations, the transcriptional activator of the operon responsible for maltose metabolism, *malT*, was the frequent target of mutations that reduced its ability

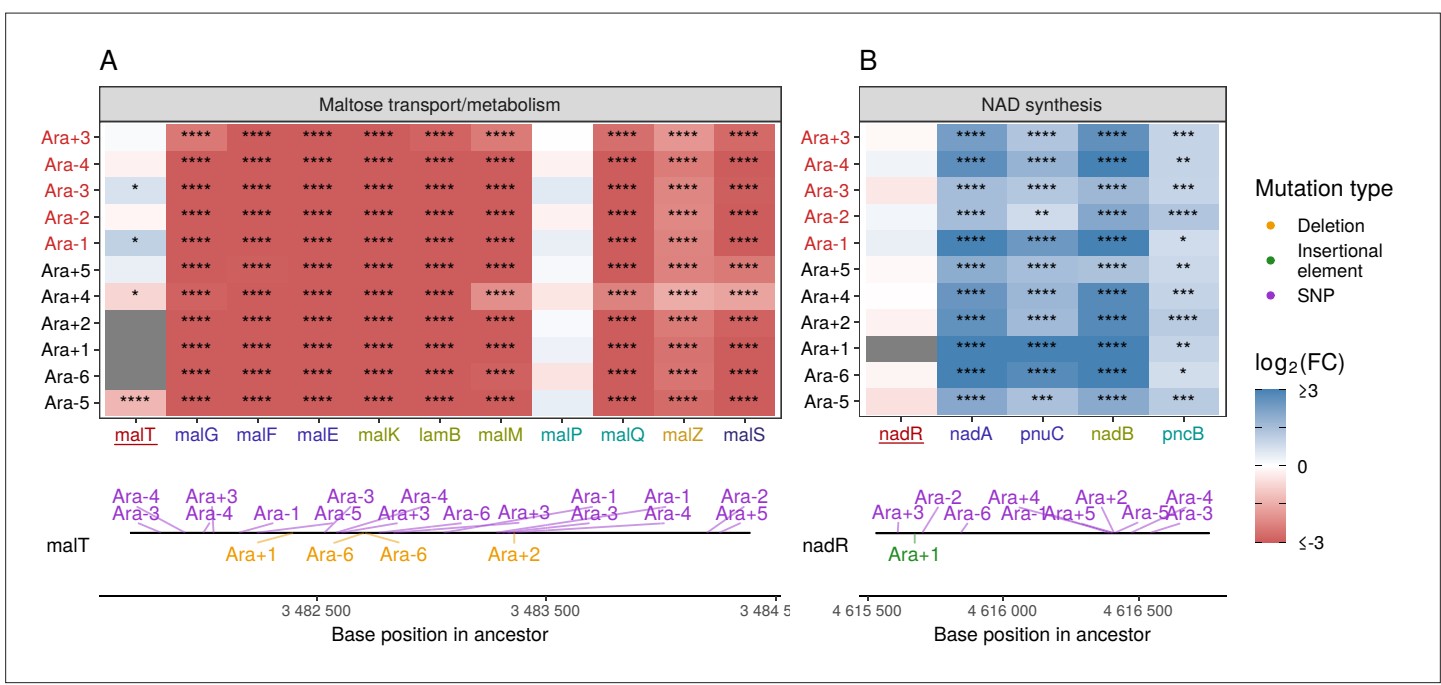

**Figure 5.** Mutations in transcriptional regulators lead to parallel changes in gene expression. RNA-seq fold-changes for genes belonging to (A) maltose-transport/metabolism and (B) nicotinamide adenine dinucleotide (NAD) biosynthesis. Gene names in each category are colored based on their operon membership. Mutations in transcriptional activator malT decrease expression of its downstream genes/operons. Mutations in transcriptional repressor nadR increase expression of its downstream genes/operons. Asterisks indicate statistical significance of fold-changes (blank: q > 0.05, *: q ≤ 0.05, **: q ≤ 0.01, ***: q ≤ 0.001 ****: q ≤ 0.0001). Gray panels in the heatmap indicate gene deletion. Lower panels show the type and location of mutations in each transcription factor.

The online version of this article includes the following figure supplement(s) for figure 5:

**Figure supplement 1.** Link between mutations and expression changes for other gene sets.

to act as a transcriptional factor, and the introduction of *malT* mutations in the ancestor had a fitness benefit (*Pelosi et al., 2006*). In *E. coli*, *malT* regulates the transcription of several operons – *malEFG* (maltose ABC transporter), *malK-lamB-malM* (*malK*, part of maltose ABC transporter; *lamB*, maltose transporter; *malM*, conserved gene of unknown function, *malPQ* (two enzymes involved in maltose metabolism), and the genes *malZ* (maltodextrin glucosidase) and *malS* (an α-amylase)). We find that each of these operons was consistently and significantly downregulated across all lines (*Figure 5A*). Changes to the *lamB* gene have also been shown to affect susceptibility to phage infection in the LTEE (*Meyer et al., 2010*).

Many categories related to the molecule nicotinamide adenine dinucleotide (NAD) appeared in our PPS (*Figure 4C*) and GO results (*Figure 4—figure supplement 2A*). In the LTEE, *nadR*, a transcriptional repressor of genes involved in NAD biosynthesis, is frequently mutated, with many mutations occurring in its DNA-binding domain (*Ostrowski et al., 2008*; *Woods et al., 2006*). All evolved clones used in this study are known to have some mutation in *nadR* (*Tenaillon et al., 2016*). Given the high frequency of parallel inactivating mutations in *nadR*, these mutations are likely adaptive as they might increase intracellular NAD concentrations leading to faster growth (*Ostrowski et al., 2008*; *Woods et al., 2006*). We find that genes directly under the regulation of *nadR* – the *nadAP* operon consisting of *nadA* (quinolinate synthase) and *pnuC* (nicotinamide riboside transporter), and genes – *nadB* (L-aspartate oxidase) and *pncB* (nicotinate phosphoribosyltransferase), were significantly upregulated in all lines (*Figure 5B*). Interestingly, four genes *nadCDEK*, which play various NAD biosynthesis roles in other pathways and are not regulated by *nadR*, were largely unaltered (*Figure 4—figure supplement 2C*). Concordantly, their transcriptional regulator, *nac*, is rarely mutated, suggesting that there is some specificity to how NAD levels may be increased in the cell.

In addition to linking the effects of specific mutations on gene expression changes in maltose and NAD regulation, we have also identified mutations that likely change the expression of genes involved in arginine biosynthesis, glyoxylate bypass system, and copper homeostasis (*Figure 5—figure supplement 1*, see Appendix A4). However, several functionally related sets of genes exist, such as flagellar assembly, sulfur homeostasis, and the glycine cleavage system – that have parallel changes in expression levels without any obvious sets of parallel mutations linking these changes (*Figure 5—figure supplement 1*). The data generated in this study will likely prove to be a rich resource for understanding the metabolic changes that occur over long periods of evolution in a simple environment such as in the LTEE, thereby adding a new dimension to the well-studied mutational changes and gene expression changes described here.

## Discussion

Adaptation to novel environments often takes unique mutational paths even when the tempo and mode of adaptation are similar across populations (*Cheng, 1998*; *Levy et al., 2015*; *Meyer et al., 2012*; *Tenaillon et al., 2012*; *Tenaillon et al., 2016*; *Therkildsen et al., 2019*). This is due, in part, to the fact that most genetic networks are highly redundant and that many mutations have pleiotropic effects. To begin to bridge the gap between parallel fitness gains in a system with mostly unique genetic changes, we wanted to study gene expression – a key link between genotype and fitness. Two key findings from our work are that (i) most of the transcriptome remains unaltered in its relative expression levels and (ii) genes with altered expression levels have remarkably similar changes (magnitude and direction of changes, pathways targeted, etc.) across all evolved lines after 50,000 generations. While parallel changes in expression profiles are perhaps not surprising given the strong selection in a well-specified environment, our work suggests that expression profiles serve as a link between the disparate mutations and similar fitness gains observed in the LTEE. Although our results do not directly implicate these parallel changes in gene expression to improved fitness, the high degree of parallelism across independently evolved populations warrants further investigation into the fitness consequences of these changes. More importantly, this suggests an optimal expression profile in any particular media that supports maximum growth. Expression profile optimization may be a mode of adaptation with each fixed mutation bringing the expression profile closer to this optimum. Nonetheless, the specific mechanisms by which the evolved lines in the LTEE have achieved similar changes in expression remain unclear. Below, we review three key proposed mechanisms that each might contribute partly to the overall story of parallelism in gene expression changes in LTEE: (i)

key regulator hypothesis, (ii) chromosomal architecture and DNA supercoiling, and (iii) growth rate-dependent changes.

## Mechanisms driving parallel expression changes

According to the 'key regulator' hypothesis, changes to one or a few genes can regulate the activity of many other genes responsible for most of the expression changes. In an earlier study of expression changes in the LTEE (*Cooper et al., 2003*), it was suggested that mutations to *spoT* observed in 8 out of 12 lines were responsible for many of the observed expression changes. *spoT* is involved in the stringent response pathways (*Traxler et al., 2008*) and regulates the activity of many genes. However, of the two lines whose expression was surveyed, Ara+1 and Ara-1, only Ara-1 contained a *spoT* mutation. When transferred to the ancestor, the Ara-1 *spoT* mutation did increase fitness by reducing the duration of the lag phase and increasing growth rates and caused similar expression changes in 11 of the 59 genes found to be altered in both Ara+1 and Ara-1. This means that other mutations in both lines were necessary to achieve changes in the remaining genes. Like *spoT*, ribosomal proteins and *rpoD* (the beta subunit of RNA polymerase) have also evolved faster than other genes in the LTEE (*Maddamsetti et al., 2017*). Mutations in these genes can have large pleiotropic effects and might contribute substantially to parallelism in observed expression changes.

DNA supercoiling is known to play a strong role in regulating transcription (*El Houdaigui et al., 2019*). All the evolved lines have mutations in genes related to chromosomal architecture, such as *fis*, topoisomerase A and B, or other genes which contribute to parallel changes in DNA superhelicity (*Crozat et al., 2010*). *Fis* was also part of the set of fast-evolving genes (*Maddamsetti et al., 2017*), suggesting that changes to chromosomal architecture are a target of selection in the system. Parallel mutations in genes affecting chromosomal architecture might also explain why we observe parallel expression changes in several pathways, such as sulfur homeostasis, despite the lack of parallel mutations in transcription factors that directly regulate them (*Figure 5—figure supplement 1D*).

While the above two mechanisms might be driving many parallel changes in expression levels, changes in the expression of some genes might simply be a consequence of faster growth. Expression levels of many genes in bacteria scale with growth rate (*Klumpp et al., 2009*; *Macklin et al., 2020*) to maintain stoichiometric concentrations. As a result, simply increasing the growth rate of replicate cultures of bacteria might produce similar expression profiles. Disentangling the effects of growth rate and genetic changes on gene expression is difficult, and therefore, we need to be cautious in over-interpreting the role of mutations in driving parallel expression changes.

## On the lack of observed translational changes

Given the universality and importance of translation to life (*Bernier et al., 2018*), it is surprising that we detect few translational changes over 50,000 generations of adaptation. Bacteria possess polycistronic genes, where many proteins are translated from a single mRNA, typically belong to the same pathway or protein complex, and are translationally regulated (*Li et al., 2014*). Therefore, it is likely that any additional translational changes to genes in an operon might disrupt the stoichiometric balance of proteins in a metabolic pathway or protein complex. It is also likely that the dynamic range of translational changes is smaller than transcriptional changes in bacteria (*Cambray et al., 2018*; *Li et al., 2014*; *Goodman et al., 2013*) or that it might take much longer than the time scales of LTEE to observe such changes.

## Conclusions

The LTEE remains a rich source for studies of evolution. Our work suggests that alterations to the global transcriptional profile is a mode of adaptation in the LTEE and that specific categories of genes have undergone similar expression changes across the lines. However, as described above, relating gene expression changes to specific mutations in LTEE is far from perfect. This is further compounded by the fact that half of the evolved lines in LTEE have a hypermutable phenotype. These genotypes have 100-fold higher mutational load than their non-mutator counterparts. It is remarkable that despite a higher mutational burden, expression patterns between mutator and non-mutator lines are highly correlated, suggesting that the bulk of the additional mutations are indeed passenger mutations (*Good et al., 2017*). While our current study has focused on expression patterns in the exponential phase, populations in the LTEE spend a significant amount of time in the stationary phase before serial

transfer. However, it remains unclear if we would observe a similar level of parallelism in the stationary growth phase or how similar the expression profiles might be across distinct growth phases. Finally, the analyses undertaken here have focused on single clones from each of the evolved lines. However, each evolved population has many distinct genotypes and segregating mutations. Taking a single-cell sequencing approach, while still challenging in bacteria (*Imdahl and Saliba, 2020*), should provide a better understanding of gene expression evolution in LTEE. Lab evolution experiments combined with high-throughput multi-level sequencing approaches offer a rich resource for studying the molecular mechanisms underlying complex adaptations and provide insights into the repeatability of evolution.

## Materials and methods

### Bacterial cell culture, recovery, and lysis

We used the following clones for generating RNA-seq and Ribo-seq datasets: Ara-1 – 11330, Ara+1 – 11392, Ara-2 – 11333, Ara+2 – 11342, Ara-3 – 11364, Ara+3 – 11345, Ara-4 – 11336, Ara+4 – 11348, Ara-5 – 11339, Ara+5 – 11367, Ara-6 – 11389, Ara+6 – 11370. Bacteria were cultured in medium as per the recipe on the LTEE website (http://myxo.css.msu.edu/ecoli/dm25liquid.html) supplemented with 4 g/L glucose instead of the typical 25 mg/L. Each culture was grown in 50 mL in a shaking incubator at 37°C at 125 rpm until an OD600 of 0.4–0.5 was reached. This took between 1.5 and 4 hr, depending on the line. Cells were recovered via vacuum filtration and immediately frozen in liquid nitrogen (LN2). Frozen pellets were stored at –80°C until lysis. A mortar and pestle were chilled to cryogenic temperatures with LN2 for lysis. The pellet was ground to a powder while submerged in LN2. Once pulverized, 650 µL of lysis buffer was added to each sample and ground further. Lysis buffer contained the following: 20 mM Tris pH 8, 10 mM $MgCl_2$, 100 mM $NH_4Cl$, 5 mM $CaCl_2$, 1 mM chloramphenicol, 0.1% v/v sodium deoxycholate, 0.4% v/v Triton X-100, 100 U/mL DNase I, 1 µL/mL SUPERase-In (Thermo Fisher Scientific AM2694). The frozen lysate was allowed to thaw until liquid, then incubated for 10 min on ice to allow complete lysis. Afterward, the lysate was centrifuged at 20,000× *g* for 10 min at 4°C, and the supernatant recovered and transferred to a new tube. Each sample was split into two for RNA-seq and Ribo-seq libraries.

### RNA-seq library preparation

Lysate destined for RNA-seq libraries was subjected to total RNA extraction using the Trizol method (Thermo Fisher Scientific 15596026) as per the manufacturer's instructions. RNA was quantified using UV spectrophotometry. We used the ERCC RNA Spike-In Mix (Thermo Fisher Scientific 4456740) in library preparation. For RNA-seq libraries, 3 µL of a 1:100 dilution of the set 1 oligos was added to the first replicate and 4 µL to the second replicate. The spike-ins were added directly to the lysate destined for RNA-seq before Trizol-based RNA extraction. Two µg of RNA with ERCC controls were subjected to fragmentation in a buffer containing final concentrations of 1 mM EDTA, 6 mM $Na_2CO_3$, and 44 mM $NaHCO_3$ in a 10 µL reaction volume for 15 min at 95°C. Five µL of loading buffer (final concentrations of 32% v/v formamide, 3.3 mM EDTA, 100 µg/mL bromophenol blue) was added to each sample, and the resulting 15 µL mixture was separated by gel electrophoresis with a 15% poly-acrylamide TBE-urea gel (Invitrogen EC68852BOX) at 200 V for 30 min. Gels were stained for 3 min with SYBR Gold (Thermo Fisher Scientific S11494), and the region corresponding to the 18–50 nucleotide fragments was excised. We excised this region so that we would have similarly sized fragments for both RNA-seq and Ribo-seq libraries. RNA was recovered from the extracted fragments by adding 400 µL a buffer containing 300 mM sodium acetate, 1 mM EDTA, and.25% w/v SDS, and freezing the samples on dry ice for 30 min. Then, samples were incubated overnight on a shaker at 22°C; 1.5 µL of GlycoBlue (Thermo Fisher Scientific AM9515) was added as a co-precipitant, followed by 500 µL of 100% isopropanol. The samples were chilled on ice for 1 hr and then centrifuged for 30 min at 20,000× *g* at 4°C. The supernatant was removed, and the pellet was allowed to air dry for 10 min. The pellet was resuspended in 5 µL of water, and 1 µL was used to check RNA concentration via UV spectrophotometry.

### Ribo-seq library preparation

Lysate destined for Ribo-seq was incubated with 1500 units of micrococcal nuclease purchased from Roche (catalog number 10107921001) and 6 µL of SUPERase-In at 25°C for 1 hr and shaken at

1400 rpm. Two µL of.5 M EGTA pH 8 was added to quench the reaction, which was then placed on ice. The reaction was centrifuged over a 900 µL sucrose cushion (final concentrations of 20 mM Tris pH 8, 10 mM $MgCl_2$, 100 mM $NH_4Cl$, 1 mM chloramphenicol, 2 mM DTT, 9 M sucrose, 20 U/mL SUPER-ase-In) using a Beckman Coulter TLA100 rotor at 70,000 rpm at 4°C for 2 hr in a 13 mm × 51 mm polycarbonate ultracentrifuge tube (Beckman Coulter 349622). The sucrose solution was removed from the tube, and the pellet was resuspended in 300 µL of Trizol, mixed by vortexing, and RNA was extracted according to the manufacturer's protocol. Samples were then separated by gel electrophoresis and purified in the same manner as for RNA-seq.

## Unified library preparation

Once fragments were obtained from RNA-seq and Ribo-seq samples, they could be subject to a unified library preparation protocol as in *Chatterji et al., 2018*; *Gupta et al., 2019*. In total, eight pooled libraries were prepared, with a final library structure of 5' adapter – 4 random bases – insert – 5 random bases – sample barcode – 3' adapter. The randomized bases function as UMIs for deduplication.

## ERCC spike-in controls and modeling

The ERCC RNA Spike-In Mix (Thermo Fisher Scientific 4456740) was used in library preparation. For RNA-seq libraries, 3 µL of a 1:100 dilution of the set 1 oligos was added to the first replicate and 4 µL to the second replicate. The spike-ins were added directly to the lysate destined for RNA-seq before Trizol-based RNA extraction. The file 'absolute_counts.Rmd' contains the code for the linear modeling using the ERCC data.

## CFU determination

Before recovery, 1 mL of culture was extracted for CFU determination. LB agar plates were used for colony growth. We performed a dilution series of that 1 mL culture from 1:10 to 1:1e6 in increments of 10; 100 µL of each dilution was spread on a plate and incubated overnight at 37°C. We determined CFU counts manually from the most appropriate dilution for each culture, usually between 1:1e3 and 1:1e6 dilutions.

## Optical microscopy

Liquid cultures were grown at 37°C with aeration, unless otherwise indicated, in DM25 medium (Davis minimal broth supplemented with glucose at a concentration of 25 mg/L) (*Lenski et al., 1991*). Before each experiment, clones were grown in liquid cultures in DM25 medium overnight at 37°C with aeration. OD600 of the cultures were 0.1–0.3. Microscope slides were prepared with 1% agarose pads, and cells were imaged by microscopy. Phase-contrast microscopy was performed using an Olympus IX81 microscope with a 100 W mercury lamp and ×100 NA 1.35 objective lens; 16-bit images were acquired with a SensiCam QE cooled charge-coupled device camera (Cooke Corp.) and IPLab version 3.7 software (Scanalytics) with 2×2 binning. Analysis of the images was performed with ImageJ (*Abràmoff et al., 2004*) and the MicrobeJ plugin (*Ducret et al., 2016*).

## Sequencing data processing

Raw sequencing data is deposited in the GEO database under the ascension GSE164308. Code for all data processing and subsequent analysis can be found in a series of R markdown documents uploaded to GitHub (https://github.com/shahlab/LTEE_gene_expression_2; *Favate, 2022*; copy archived at swh:1:rev:b8fd5632d258bc78ae136208ef1ad1fe6d359483). The file titled 'data_processing.Rmd' contains the code for processing the raw sequencing data. Briefly, the following tools were used to remove adapters (cutadapt, *Martin, 2011*), deduplicate (BBtools dedupe.sh script), and demultiplex (FASTX-toolkit barcode splitter script) the data. Only reads of at least 24 nucleotides in length after trimming were retained for alignment. Transcript quantification for both sequencing-type datasets was performed with kallisto (*Bray et al., 2016*). hisat2 (*Kim et al., 2019*) was used to align Ribo-seq data for analyzing changes at specific codons. For this analysis, alignment was performed against a custom transcriptome that padded each coding region with 25 nt on the 3' and 5' ends to allow for better mapping of ribosomes at the start and stop codons.

## Differential expression analysis of gene expression

Code for this section can be found in the file 'DEseq2.Rmd'. We used DEseq2 (*Love et al., 2014*) with the 'apeglm' normalization (*Zhu et al., 2019*) for differential expression. In estimating fold-changes, we compared the four replicates of the ancestors (two each from ancestors of Ara+ and Ara-) to two replicates of each of the evolved lines. Because some genes in some lines contained indels or were deleted entirely, some transcripts were missing from the transcriptome fastas used to create indices for alignment. We added these genes back to Kallisto's counts with estimated counts of 0 and assigned them fold-changes of NA. Count matrices containing identical complements of transcripts were used in the differential expression analysis for each line, such that all evolved lines had the same complement of genes as the ancestors.

## Change in ribosomal density analysis

We used Riborex (*Li et al., 2017*) to analyze changes in ribosomal density. The same count matrices used for DEseq2 were used here, and comparisons were made in the same manner of four ancestral samples (two lines, two replicates each) to two evolved clones (one line, two replicates). The code for this section can be found in the file 'riborex.Rmd'.

## Linear mixed modeling for changes in ribosome density

Code for this section can be found in 'fig_3.Rmd' under the 'Modeling' heading. Briefly, we fit linear mixed models using the 'lme' function from the R package 'nlme' to test if stop codons showed a larger decrease in ribosome densities (relative to the ancestor) as compared to the sense codons. Briefly, linear mixed models perform linear regression allowing for fixed effects (i.e., a population-level effect) and potential random effects (i.e., effects restricted to pre-specified subpopulations of the data). In this case, the random effects correspond to evolved line-specific effects on log2 ribosome density fold-changes. We fit various linear mixed models allowing for different constraints on the random effect slopes and intercepts, as well as an ordinary linear regression (i.e., no random effects across evolved lines) as the null model. Models were compared using the Akaike information criterion (AIC): the model with the lowest AIC score is generally considered the best model. Although we identified three linear mixed model fits that had similar performance based on the AIC score (i.e., the difference in AIC scores was less than 2), we chose to use the simplest model, which allowed for uncorrelated random effect intercepts and slopes. This model also happened to be the model with the lowest AIC score. For comparison, this model was approximately 27 AIC units better than the ordinary linear regression.

## Codon-specific positioning of Ribo-seq data

Code for this section can be found in the file 'codon_specific_densities.Rmd'. It has been shown that mapping bacterial Ribo-seq reads by their 3' ends is more accurate than 5' mapping (*Mohammad et al., 2019*), so we mapped the A-site position of a read by using a fixed offset of 37 nt (12 nt offset+25 nt addition to transcript ends). To calculate ribosome densities on a codon for a gene, the number of reads mapping to a codon was normalized to the total number of reads mapping to that gene in a replicate and line-specific manner. Genome-wide codon density is calculated by taking genes with at least 100 reads mapping to them and taking the average number of normalized reads mapping to each codon across that set of genes as the genome-wide codon density. Three nucleotide periodicity is determined in the file '3nt_periodicity.Rmd'.

## Functional analysis

We used three different functional analysis methods – GO (using the R package topGO), KEGG (using the R package clusterprofiler; *Yu et al., 2012*), and PPS (*Karp et al., 2019*). The code for each of these analyses can be found in the Rmd files named 'go.Rmd', 'kegg_analysis.Rmd', and 'manual_PPS.Rmd', respectively. PPS are calculated as follows: each pathway is composed of at least one reaction, and each reaction is completed by at least one enzyme. First, a reaction perturbation score is calculated for each reaction in a pathway, defined as the absolute value of the largest fold-change of an enzyme associated with that reaction. To calculate PPS, for a pathway having $N$ reactions, PPS $= \mathrm{sqrt}((\Sigma RPS^2)/N)$. Additionally, a document titled 'kegg_sensitivity.Rmd' tests the effects of adding deletions to our analysis.

## Acknowledgements

We thank Richard Lenski for generously providing clones from ancestral and 50,000 generations of the LTEE. We thank Olivier Tenaillon and Richard Lenski for helpful discussions. PS is supported by NIH/NIGMS grant R35 GM124976, NSF DBI 1936046, subcontracts from NIH/NIDDK R01 DK056645, R01 DK109714, and R01 DK124369, as well as start-up funds from the Human Genetics Institute of New Jersey at Rutgers University. ALC is supported by the INSPIRE (IRACDA New Jersey/New York for Science Partnerships in Research and Education) Postdoctoral Program (NIH PAR-19–366). SSY is supported by start-up funds from the Waksman Institute and Rutgers University.

## Additional information

### Competing interests

Premal Shah: is a scientific advisory board member of Trestle Biosciences and consults for Ribo-Therapeutics. Is also a director at an RNA-therapeutics startup. The other authors declare that no competing interests exist.

### Funding

| Funder | Grant reference number | Author |
|---|---|---|
| National Institute of General Medical Sciences | ESI-MIRA R35 GM124976 | Premal Shah |
| National Science Foundation | DBI 1936046 | Premal Shah |
| Rutgers, The State University of New Jersey | Start-up funds | Srujana S Yadavalli |
| National Institutes of Health | IRACDA NJ/NY for Science Partnerships in Research and Education Postdoctoral program NIH PAR-19-366 | Alexander L Cope |
| National Institute of Diabetes and Digestive and Kidney Diseases | Subcontract from R01 DK056645 | Premal Shah |
| National Institute of Diabetes and Digestive and Kidney Diseases | Subcontract from R01 DK109714 | Premal Shah |
| National Institute of Diabetes and Digestive and Kidney Diseases | Subcontract from R01 DK124369 | Premal Shah |

The funders had no role in study design, data collection and interpretation, or the decision to submit the work for publication.

### Author contributions

John S Favate, Resources, Data curation, Software, Formal analysis, Validation, Investigation, Visualization, Methodology, Writing - original draft, Writing – review and editing; Shun Liang, Investigation, Methodology; Alexander L Cope, Srujana S Yadavalli, Investigation, Writing – review and editing; Premal Shah, Conceptualization, Resources, Software, Formal analysis, Supervision, Funding acquisition, Investigation, Methodology, Project administration, Writing – review and editing

### Author ORCIDs

John S Favate http://orcid.org/0000-0001-6344-4854
Alexander L Cope http://orcid.org/0000-0002-5756-7812
Premal Shah http://orcid.org/0000-0002-8424-4218

### Decision letter and Author response

Decision letter https://doi.org/10.7554/eLife.81979.sa1

Author response https://doi.org/10.7554/eLife.81979.sa2

## Additional files

### Supplementary files

• Supplementary file 1. Results of the kallisto alignment for all samples. Counts in this file were first rounded, and new transcripts per million (TPM) were calculated based on rounded counts. This file was generated using 'data_cleaning.Rmd' (https://github.com/shahlab/LTEE_gene_expression_2/tree/main/code/data_processing).

• Supplementary file 2. Results from DESeq2 for all samples. Generated from 'DESeq2.Rmd' (https://github.com/shahlab/LTEE_gene_expression_2/tree/main/code/analysis).

• Supplementary file 3. Quantifications from our optical microscopy. This table is supplied and is not generated from the code.

• Supplementary file 4. Our colony-forming unit (CFU) numbers. This table is supplied and is not generated from the code.

• Supplementary file 5. Amounts of ERCC spike-ins added to each sample and their abundance in the sequencing libraries. This table is supplied and is not generated from the code.

• Supplementary file 6. Measures of mRNA abundance per colony-forming unit (CFU). Generated from 'absolute_counts.Rmd' (https://github.com/shahlab/LTEE_gene_expression_2/tree/main/code/analysis).

• Supplementary file 7. Results from riborex. Generated from 'riborex.Rmd' (https://github.com/shahlab/LTEE_gene_expression_2/tree/main/code/analysis).

• Supplementary file 8. Calculated genome-wide codon densities. Generated from 'codon_specific_densities.Rmd' (https://github.com/shahlab/LTEE_gene_expression_2/tree/main/code/analysis).

• Supplementary file 9. KEGG search results. Generated from 'kegg_analysis.Rmd' (https://github.com/shahlab/LTEE_gene_expression_2/tree/main/code/analysis).

• Supplementary file 10. GO search results. Generated from 'go.Rmd' (https://github.com/shahlab/LTEE_gene_expression_2/tree/main/code/analysis).

• Supplementary file 11. Pathway perturbation score (PPS) calculations. Generated from 'manual_pps.Rmd' (https://github.com/shahlab/LTEE_gene_expression_2/tree/main/code/analysis).

• Supplementary file 12. Mmutation data for our clones as downloaded from https://barricklab.org/shiny/LTEE-Ecoli/. This file is supplied and not generated from the code or can be downloaded from the website.

• MDAR checklist

### Data availability

Sequencing data have been deposited in GEO under accession code GSE164308. All data generated or analyzed during this study are included in the manuscript and supporting file; Source Data files have been provided for all figures. Code for all data processing and subsequent analysis can be found in a series of R markdown documents uploaded to GitHub https://github.com/shahlab/LTEE_gene_expression_2 (copy archived at swh:1:rev:b8fd5632d258bc78ae136208ef1ad1fe6d359483).

The following dataset was generated:

| Author(s) | Year | Dataset title | Dataset URL | Database and Identifier |
|---|---|---|---|---|
| Favate J, Liang S, Yadavali S, Shah P | 2022 | Landscape of transcriptional and translational changes over 22 years of bacterial adaptation | http://www.ncbi.nlm.nih.gov/geo/query/acc.cgi?acc=GSE164308 | NCBI Gene Expression Omnibus, GSE164308 |

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

# Appendix 1

## A1. Determination of Ara-2 ecotype

Analysis for determination of Ara-2 ecotype can be found in the file 'araM2_ecotype.Rmd'. Briefly, we compared mutations in our clones to the mutations determined in *Plucain et al., 2014*. Our clone of Ara-2 does not possess mutations in the arcA or gntR genes. We also compared mutations in our clone against the list of mutations unique to the S or L ecotype and found that our clone possesses many mutations unique to the L type but not the S type. Finally, *Le Gac et al., 2012*, found two large 35 and 41 kilobase deletions in the S lineage at 40,000 generations, neither of which are present in our clone at 50,000 generations.

## A2. The potential effects of increased sugar in the culture medium

The LTEE media recipe uses 25 mg/L glucose. However, this low glucose environment leads to low cell densities and constrains our ability to generate matched RNA-seq and Ribo-seq samples with sufficient depth to perform genome-wide analyses from the same culture. To overcome limitations of cell densities, we used 4 g/L, the amount of sugar specified in the agar recipe used for solid growth assays on the LTEE website (http://myxo.css.msu.edu/ecoli/dmagar.html). The increased glucose level in our medium is expected to affect the final cell density rather than the growth rate during the exponential phase. Additionally, though our experiment takes place 30,000 generations after the *Cooper et al., 2003*, study, we observe similar patterns in expression changes (*Figure 1—figure supplement 3A*). This suggests that some patterns may have reached fixation long ago and that bacteria may behave similarly across the two experiments. Finally, even in the case where the increased glucose has altered the physiology of cells in our cultures, the fact that we see parallel patterns of differential expression relative to the ancestor in each evolved line indicates that we are observing heritable differences from the ancestor.

## A3. Absolute abundances and CFU counts

We used CFUs of our cultures as a measure of cell densities to generate each library. However, filamentation of cells in our cultures can bias our estimates of cell densities since it remains unclear whether a colony was initiated from a single cell or a filament. In our data, volume increases are best correlated with length or aspect ratio as opposed to width (*Figure 2—figure supplement 1C*). This suggests that while some volume increases are truly individual cells getting larger, exceptionally large cells are likely chains. In the absence of absolute changes, simply undercounting the number of cells would also produce the observed results. Removal of large, presumably filamentous cells using the same filtering metric as in *Grant et al., 2021* (0.21 fL ≤ volume ≤ 5.66 fL, *Figure 2—figure supplement 1B*) has little effect on our median cell volumes and hence does not affect results that use the median volume, such as those in *Figure 2E*. That said, the amount of transcripts estimated from our data is well over what is believed to be present inside a bacterium (*Moran et al., 2013*), so CFUs likely underrepresent the number of cells used to make each library. Moreover, a CFU assay only considers living cells, whereas dead cells, depending on their time of death relative to collection time, could also contribute to RNA abundance but not CFUs.

## A4. Analysis of altered pathways

Flagellar assembly was the top category in the KEGG results, and categories relating to motility or flagella were frequent in the PPS and GO analyses. Flagella are used for motility and allow bacteria to move to new environments when necessary. Downregulation of flagellar genes is a common adaptation in laboratory-based evolution experiments (*Edwards et al., 2002*) and was a principle finding in *Cooper et al., 2003*. We also observed downregulation of the *flgBCDEFGHIJK*, *flgAMN*, and *flhABE* operons in all but one evolved line (*Figure 5—figure supplement 1A*, upper panel). These operons contribute various proteins to the flagellar apparatus and are regulated in part by the transcription factors *flhC* and *flhD*, which themselves have complicated regulation dictated by various environmental factors (*Soutourina and Bertin, 2003*). *flhC* and *flhD* are downregulated in three of the evolved lines but mostly unaltered in the others. These genes are rarely mutated in the clones used in this study (*Figure 5—figure supplement 1A*, lower panel). Because *E. coli* B is thought to be non-motile (*Jeong et al., 2009*), it's likely that the downregulation of these genes is due to the removal of an unnecessary function and was fixed early on in the experiment. The lack of parallel changes in transcriptional regulators *flhCD* suggests that other mechanisms may play a part in causing the downregulation of these genes.

Terms relating to arginine and other amino acids were common in our results. We found that genes related to arginine synthesis were statistically significant and upregulated in many lines (*Figure 5—figure supplement 1B*). Upregulation of genes in amino acid synthesis pathways could increase intracellular amino acid amounts, allowing faster translation and leading to faster growth. Alternatively, the arginine synthesis pathways have many intermediate molecules which can be fed into other metabolic pathways, one of which could also allow faster growth. *argR*, which represses transcription of these genes when L-arginine is abundant (*Caldara et al., 2006*), frequently contains mutations in or around its coding sequence and is unaltered in its expression. As such, some of these mutations may have disabled the repressive ability of *argR*, leading to the increased expression we observe here.

The glyoxylate bypass system allows *E. coli* to utilize acetate as a carbon source. It is composed of the *aceBAK* operon and is regulated by *iclR* and *arcAB* (*Okamura-Ikeda et al., 1993*). Acetate is a metabolic by-product but can be returned to central carbon metabolism for biosynthetic reactions by this system. Previous studies have shown that mutations in *iclR* and *arcB* cause derepression of their target genes are beneficial in the LTEE (*Quandt et al., 2015*). Consistent with these results, we found that the *aceBAK* operon was upregulated in 9 of 11 evolved lines (*Figure 5—figure supplement 1C*). This confirms the hypothesis from *Quandt et al., 2015*, that mutations to *iclR* and *arcB* derepress enzymes involved in acetate metabolism.

Sulfur is a critical component of many biological molecules, like amino acids, and participates in creating other structures like iron-sulfur cluster proteins. Organic sulfur is transported across the cell membrane by proteins from the *cysPUWAM* operon, which encodes for a sulfate/thiosulfate importer (*Sirko et al., 1995*), the *gsiABCD* operon which encodes for a glutathione importer (*Suzuki et al., 2005*), the *tauABCD* operon which codes for a taurine importer (*Eichhorn et al., 2000*), and *tcyP*, the major L-cysteine importer (*Chonoles Imlay et al., 2015*). We found that many of these genes were downregulated in many lines (*Figure 5—figure supplement 1E*). The *cysB* gene positively regulates these genes and was downregulated in most lines and contained few mutations. The sources of organic sulfur in the medium used in the LTEE are ammonium and magnesium sulfate, for which the *cysPUWAM* operon functions as the importer. The mechanism and reasons for alterations to these operons remain unclear. One hypothesis is that the amount of organic sulfur in the medium is sufficient to allow the downregulation of sulfur transport systems without impacting downstream pathways that require sulfur and negatively impacting growth, thus saving energy by not transcribing or translating them.

Glycine plays a role in protein construction and can be a building block for other metabolic pathways such as one-carbon metabolism or serine synthesis (*Okamura-Ikeda et al., 1993*; *Wilson et al., 1993*). We found that the *gcvTHP* operon, which encodes for proteins in the glycine cleavage system, was upregulated in many of the evolved lines. Increases in the levels of compounds involved in this set of reactions may directly increase growth rates. Though some mutations exist in and around transcriptional regulators of these genes, their effects are unclear. Whether changes to these genes are due to changes in their transcription factors or other changes, the upregulation of these genes in many lines suggests that it may be beneficial.

Copper and silver have antibacterial properties (*Ingle et al., 2014*), and bacteria have evolved systems to mitigate toxicity from these elements. The *cusCFBA* operon, regulated by the *cusRS* sensor kinase, codes for proteins that transport copper and silver ions out of the cell (*Nies, 2003*). Additionally, the cytoplasmic copper chaperone *copA*, regulated by *cueR* (*Meydan et al., 2017*), and *cueO* (multicopper oxidase; *Grass and Rensing, 2001*) regulate copper homeostasis in the cell. These genes contained deletions in five of our clones and were downregulated in three of the six lines where they remained (*Figure 5—figure supplement 1F*). Overall, eight of the eleven lines surveyed here had defects in these systems. This suggests that these genes may be selected for removal or downregulation. In contrast to natural environments, the laboratory environment is likely free of copper and silver, rendering these systems dispensable. That said, because many of these genes are casualties of large deletions, it's not obvious which genes, if any, provide a fitness benefit in the system.

