## [Editor Report]

This paper comprehensively analyzes how gene expression has changed in eleven *E. coli* strains after 50,000 generations of laboratory evolution. It confirms that, overall, changes in RNA levels are more reproducible than the underlying genetic changes and begins to investigate how some of these changes lead to increased fitness in this environment. This dataset will be a valuable resource for testing theories about how genotypic and phenotypic evolution are coupled and for understanding how bacterial gene regulatory networks evolve during adaptation.

---

## [Decision Letter]

**Decision letter after peer review:**

[Editors’ note: the authors submitted for reconsideration following the decision after peer review. What follows is the decision letter after the first round of review.]

Thank you for choosing to send your work entitled "The landscape of transcriptional and translational changes over 22 years of bacterial adaptation" for consideration at *eLife*. Your initial submission has been assessed by a Senior Editor in consultation with a member of the Board of Reviewing Editors. Although the work is of interest, we are not convinced that the findings presented have the potential significance that we require for publication in *eLife*.

Specifically, this work describes expression changes that have occurred in long-term evolution lines. The data are interesting and the analyses well-performed. As you will see in the reviewers' comments below, they anticipate that this work will be an excellent resource for the field, but also felt that the manuscript failed to provide the advance in biological insight required for publication in this particular journal. I think these three reviews, collectively, do a good job of articulating the primary concerns. I'll also note that in the post-review discussion, even the reviewer with the most positive originally submitted review (shown below) agreed with concerns raised by the other reviewers in their individual comments. At the same time, we all agreed there was more potential in these data than realized in the current manuscript, so would be happy to consider a substantially revised paper that focuses more clearly on biological insights that can be comprehensively and cleanly addressed with these data. Determining which mutation(s) are driving the expression changes was also identified as something that would increase the impact of this work.

*Reviewer #1 (Recommendations for the authors):*

The authors present a detailed analysis of transcriptional (RNA-seq) and translational (Ribo-seq) changes occurring during the LTEE. The manuscript is generally well written and organized, and analyses seem to be appropriate. I have two general comments. First, it would help for the large amount of data to be presented with more focus to the motivating biological question. Second, which of the observed changes are cause, and which consequence, of changes in fitness.

1. I found it hard to pin down the overarching goal of the work. A lot of data is presented, and it would be helpful for readers to understand the motivating questions that are being addressed. There are certainly many candidates. For example, 'bridging the gap between disparate genomic changes and parallel fitness gains' (L14), '[characterization of] the mechanistic basis of adaptation' (L39), '[exploration of] the role that transcription and translation play in increasing growth rates' (45). To me, some points seem difficult to pin down (what does 'bridging the gap' really entail?) while others oversell actual results (relatively little mechanism is presented, and, as far as I can see, nothing that directly connects any observed molecular phenotype change to a change in fitness/growth rate). I think the Introduction, and the manuscript generally, would be improved by identifying and presenting clear goals of the work, ideally related to significant biological questions that the work allows to be addressed. It is telling that the Discussion is very short, presenting almost no relationship between results presented here and previous work.

2. Many bacterial genes have growth rate-dependent expression – due, for example, to effects of cell growth on regulator concentrations (e.g., Cell 2009 139:1366). Such dependence will tend to create parallel gene expression changes in faster growing strains relative to a slower growing reference. Clearly, this effect doesn't explain all the changes observed in this study, but it is not clear how many it does explain, and, perhaps more importantly, how to interpret the possibility that a fraction of all expression changes are an effect rather than a cause of fitness increases. At minimum, I'd like the authors to present and discuss this point generally and, where appropriate, to discuss how it effects the specific conclusions they make (and the kind of questions that can reasonably be asked – e.g., growth rate dependent expression changes make it much more difficult to work back from expression changes to causal mutations).

*Reviewer #2 (Recommendations for the authors):*

This paper describes a systems biology study of how RNA expression levels and mRNA translation rates (ribosomal occupancy) changed as eleven *E. coli* populations adapted to laboratory conditions over 50,000 generations during the Lenski long-term evolution experiment (LTEE). The main finding is that the molecular phenotype-that is, which genes had altered expression and by how much-of the eleven independently evolved bacterial strains was more conserved than the underlying genetic changes-that is, which genes directly sustained mutations in the lineage leading to that strain during adaptation. Nearly all changes were at the level of transcriptional control of gene expression, with very few changes in how mRNAs were translated. These findings should be of broad interest because they inform theories about how gene expression evolves during adaptation.

Strengths

This study combines RNA-Seq and Ribo-Seq to look at changes in RNA abundance and mRNA translation rates. It represents a substantial advance, both in the techniques used and the scope in terms number of *E. coli* strains analyzed, relative to prior work that had analyzed two strains at 20,000 generations using an array-based technology close to twenty years ago. Other positive aspects of the study are that it uses spike-in controls for determining absolute RNA abundance and makes an effort to account for changes in the sizes of evolved cells. Changes in gene regulation are examined at the level of cellular processes but also profiled for specific regulons (e.g., malT and nadR). The finding of an apparent increase in the rate of translation termination is novel. Overall, the methods display a high degree of rigor. Finally, the citation and description of past results from the LTEE is thorough and appropriate. The authors should be commended for this, since this is a rather large amount of history to take into consideration, and they do not appear to have worked on this system before.

Weaknesses

There are some potential limitations/caveats related to the exact growth conditions used by the authors, which have a much higher concentration of the limiting nutrient (glucose) than was present during evolution of these strains in the LTEE. Some interpretation and analysis of how gene expression changes are related to fitness evolution could be improved. The correlation between large deletions and reduced gene expression of the genes contained within them could be examined in a different way that might lead to a significant effect. Inferences about the importance of how increases versus decreases in gene expression contribute to fitness evolution are indirect and do not appear to be completely justified.

1) Given that this was a major result, we felt that the paper could be improved by including a simple figure illustrating the similarity of the evolved strains to one another compared to how different they all are from the ancestor. We believe it would be easy to show this using a PCA plot. It is also possible that this analysis would show that some of the evolved lines are more like one another than others are, which may be interesting in light of some of our other recommendations.

2) The manuscript largely assumes that fitness is the same across the eleven LTEE populations. While it is true that the fitness improvements in each population are very similar, both the Wiser et al. 2013 Science paper cited in the Introduction and the follow-up Lenski et al. 2015 Proc Royal Soc B paper, do show that there are also systematic differences between some populations. For example, the hypermutators do tend to have higher fitness than the others, on average. There is also the case of population Ara+1, which appears to be lagging in fitness because it sustained an unusually large number of transposon-mediated mutations (Consuegra et al. 2021 Nat Comm). Eleven clones are probably not enough to try to start predicting fitness from gene expression profiles, but it would be interesting if any global analysis of the data found that outliers in terms of gene expression were also outliers in terms of fitness.

3) We think it is important to change the plotted per CFU values in Figure 1 to be per typical single cell volume. The filamentation observed in several of the evolved lineages dramatically affects the estimates of RNA abundance per CFU. While this is factually correct according to the methods and noted in a supplementary analysis paragraph, it would be much better to correct Figure 1 to use a different basis than CFUs, so that it does not give the impression that the per cell mRNA levels changed by >10-fold for some evolved strains. We believe the authors could correct this to be "per evolved cell volume", by estimating how many typical cell equivalents there are on average per CFU (filamented or not).

4) The observation that downregulated versus upregulated genes are more likely to show the same change in other lines is interesting (Line 208-212). However, we don't understand why these results indicate that there are "fewer genes and pathways whose downregulation increases fitness" necessarily. What is the connection to fitness?

5) The test of the hypothesis that genes that were deleted in some lineages would be downregulated in other lineages in which they were not deleted gave a negative result. As the authors suggest, it may be that downregulation of just one of the deleted genes yields the fitness benefit for the entire deletion. In addition to the current analysis, we would recommend repeating the analysis in a way that tests this refined hypothesis that at least one deleted gene is downregulated in the other lines. It may be possible to identify which gene or genes "drove" the deletion and which genes were collateral deletions.

6) *E. coli* cells used for RNA-Seq and Ribo-Seq were cultured in a slightly different medium than was used for the evolution experiment. The base media is the same, but a much higher concentrations of glucose was used (4,000 µg/mL versus 25 µg/mL). Presumably, this was necessary in order to be able to harvest enough cells for the RNAseq and ribosomal profiling experiment. Still, this difference should be noted and any affect that it might have on interpreting the data should be discussed. There also appears to be another minor difference in that the Lenski website recipe calls for thiamine supplementation.

7) There are some deep genetic divergences and large phenotypic changes in some of the LTEE populations that make it important to know which type of clonal isolate was analyzed here to interpret the results. Most importantly, population Ara-3 evolved citrate utilization, which enables it to grow to a higher cell density. Is the clone that was analyzed Cit+ or Cit-? Also, population Ara-2 diverged into "large" and "small" colony types. What is the type of the clone that was analyzed from each of these populations? If a Cit+ clone was used, were cells harvested at an early enough point that gene expression reflects growth of these cells on glucose (or a glucose/citrate mixture) rather than solely on citrate?

8) The Methods section should be revised. Currently, the quality and level of detail is very uneven. There are placeholders and mixed citation styles that make it look like some of this section was still in rough draft form. Certain sections may give too much information. For example, the RNAseq library preparation methods seem to be exactly from a standard NEBnext kit? It may be better to state the differences from the standard protocol, if any, in this section. Other sections seem to leave out important information. The cell size section discusses measuring the length of the cells, but the Results on line 98 focus on cell volume, not length. As another example, perhaps too much detail is given in terms of the gel run time on Line 537, but the key detail of what the fragment size range of the "region corresponding to the expected product size" that was excised from this gel is not provided on Line 538.

9) Several of the earlier Results subsections mention that a result from the RNA-Seq data was similar for the Ribo-Seq data before the Ribo-Seq data is fully described. It may be best to wait on making those comparisons and consolidate all of those statements under the Ribo-Seq section.

10) Line 131: Define TPM in the main text here. Currently, it is only defined in the figure legend.

11) Line 140: This statement appears to have a typo that results in an incorrect meaning: "each of the lines was founded on a unique set of mutations". Perhaps they mean that each of the lines accumulated a unique set of mutations during the LTEE?

12) Line 180: This result is in Figure 2C instead of 2B.

13) Line 610: This should read "two samples of each evolved clone" rather than "2 evolved clones".

*Reviewer #3 (Recommendations for the authors):*

The LTEE holds a special place in the history of evolutionary biology and there is value in learning more about this classic case study. As an evolutionary biologist (not a microbiologist), my primary interest is what does this study of expression add to our understanding of evolution within the LTEE. This work is descriptive rather than testing well-motivated hypotheses but the authors have unearthed some intriguing patterns.

1) Much of the emphasis in this work is on the extent to which changes in expression are parallel. I have several concerns on this front.

a) What is the expectation about the degree of parallelism? In several places the authors refer to there being a "high" degree of parallelism. "High" compared to what? Zero? Is their null expectation for populations evolving in identical environments to have zero parallel expression changes? In Figure 3, they make a comparison of shared expression changes to shared genetic changes. Of course, a single genetic change (for example, in a transcription factor) could cause expression changes in many downstream genes so should not a higher number of shared expression changes than genetic changes be expected? I find it bothersome to be told that there is a "high" degree of parallelism when there is no expectation. Was it possible to observe 5 times more parallelism than they observed? Compared to that possibility, they observed a "low" degree of parallelism.

b) A practical issue with parallelism is statistical power. Though statistical details are annoyingly scant throughout, it appears the authors have typically required a gene to exhibit "significant" evolutionary change in two (or more) lines to be considered parallel. Because statistical power to detect change in any one line is less than 100% (and for many changes will probably be more like 10%), the power to observe parallelism will be limited, perhaps severely, in many cases. Unfortunately, I have no sense of the extent to which parallelism in this study is underestimated because of this problem.

So what are we left with? In many cases throughout the manuscript we can say there is some parallelism. I do not find the observation of some degree of parallelism particularly interesting or surprising, especially given what we already know about the LTEE. To me, the question is whether the degree of parallelism meets expectation or is remarkable and that question is not addressed.

2) The negative correlation between evolved change and ancestral expression level.

This pattern shown in Figure 2C and D and Figure S6 is surprising to me. To me, this was one of the most important results and should have been more prominently featured. My a priori expectation would be no correlation, yet they observe this negative correlation in every line (Figure S6). The authors hypothesize that this may be because of a biophysical constraint ("maximally expressed" genes can only evolve lower expression and "minimally expressed" genes can only evolve increased expression). That hypothesis should certainly be considered. I have no better hypothesis but I find it surprising if there are enough genes close to these limits to drive this pattern. From Figure S6 it does not appear this pattern is driven by genes at the range limits of ancestral expression. Is there work in *E. coli* about whether a reasonable fraction of genes are at an expression limit? (For genes at the high expression limit, presumably gene duplication can occur as a means to increase expression. I realize such mutations would be rare relative to point mutations and small indels.) The reported pattern would be even more compelling if they had some additional means to evaluate the biophysical constraint hypothesis.

3) Parallelism of up- vs. down-regulated expression

Though there are problems with expectations for the extent of "parallelism", this issue is avoided in some cases. One intriguing case is for parallelism in genes that evolve up- vs. down-regulated expression. The authors report that "more downregulations were shared across lines than upregulations". To me, this was one of the most interesting patterns they reported because there is a simple expectation of equality between up and down regulation. However, I have several concerns.

a) It wasn't clear to me there was any statistical test of this claim.

b) It seems that one should control for the number of down vs- up regulated genes (i.e., if there are 500 down regulated genes and 200 up-regulated ones in Line 1, then it would not be particularly interesting we found that Line 2 shared 50 of the down-regulated ones but only 20 of the up-regulated ones

c) The authors also report that down-regulated changes tend to larger in magnitude than up-regulated ones. This is a key piece of information. With respect to the power comment above (1b) this means that there will be greater power to detect parallel changes for up-regulated vs. down-regulated changes.

I suggest using a statistical model that attempts to account for these issues.

4) Importance of transcription vs translation to expression evolution

a) The authors find highly correlated values between RNAseq and Riboseq data. This is not at all surprising. Such correlations at the genome level are not useful because the among-gene variation in expression is so large (I have a similar complaint with lines 150-153).

b) The authors use Riborex to test for evolved differences in ribosomal densities and find very few changes. They use this as a basis for arguing that transcription is far more important than translation for expression evolution. They may well be correct. However, I suspect that the statistical power to detect changes in ribosomal density is much lower than for detecting transcription changes (e.g., ribosomal density changes are hampered by measurement error in BOTH RNAseq and RIBOseq data). Given that the power to detect ribosomal density changes is almost certainly much lower than for transcription changes, it seems premature to make much of a claim about transcription vs. translation.

5) Faster translation termination

Figure 4C is quite dramatic. That is a very interesting result.

a) This another example where too few information is provided. What is the statistical test behind the p-value shown? Based on the figure, there is a highly significant difference but I suspect the test they did was wrong. Did they run a model that accounts for both "line" and the same codon represented in every line? I suspect they did a two-sample t-test and thus have an inflated degrees of freedom.

b) Naively, I would have thought there would always be selection favouring faster translation termination (not just in the LTEE). Is there some plausible reason for why there should selection for faster translation termination in the LTEE than in the many millions of years prior to the start of the experiment?

In summary, this is a descriptive study. While this work adds to our understanding of the LTEE, its importance for evolutionary biology is less clear. The impact of this manuscript will be determined by the authors making clear what they view as the most important patterns and the interpretation of those patterns.

With respect to my comment 3, here is a potential way to improve these issues: Using all genes that have evolved significantly in at least 1 line, run something like the following statistical model:

model <- glm(cbind(Nsig, 11 – Nsig) ~ UpOrDown + Magnitude + AncExpression, family = binomial)

where Nsig is the number of lines where it significant

UpOrDown is an indicator whether expression evolved up or down

Magnitude is the |Log2FC| averaged over only the lines where it evolved significantly (because those are the lines that determined the genes inclusion in the set for this analysis)

AncExpression is the Log(TPM) in the ancestor.

The last two terms are an attempt to control for power to detect parallelism. It won't be perfect but it will be a considerable improvement over the current version.

Figure 2C. "…only statistically significant" genes. Significant in what comparison? In any 1 line vs. ancestor?

[Editors’ note: further revisions were suggested prior to acceptance, as described below.]

Thank you for resubmitting your work entitled "The landscape of transcriptional and translational changes over 22 years of bacterial adaptation" for further consideration by *eLife*. Your revised article has been evaluated by Detlef Weigel (Senior Editor) and a Reviewing Editor.

The manuscript has been improved but there are some remaining issues that need to be addressed, as outlined below:

As you will see from the comments below, the reviewers were largely happy with the significant revisions you have provided. I agree with reviewer 1 that the regressions are problematic, as you are underpowered for genes with low expression. Please do add the PCA, as suggested by Reviewer 3.

*Reviewer #1 (Recommendations for the authors):*

The current draft of this article is improved, though it still feels a little disjointed and meandering. It lacks the razor focus that takes one from a research question in the introduction, to the most relevant results addressing that question, and finally to a discussion of those results in the context of the field and other studies. That said, these are very interesting and complex datasets. The article communicates a number of interesting findings clearly, and it makes the datasets available so that others can continue to analyze them.

1) I found the paragraph beginning on Line 146 describing the negative relationship between fold-change in evolved gene expression and gene expression level in the ancestor to be problematic in two ways.

First, I worry about how possible limitations in what types of expression changes can be detected may affect the regressions in Figure 1—figure supplement 3b. For highly expressed genes, I don't doubt that there is more downregulation than upregulation. But, for lowly expressed genes, a smaller initial number of counts will limit how large of a negative fold-change could be observed at some point and whether such a change can be judged as statistically significant (and therefore included in the regression). It seems like a non-negligible number of genes only have <10 counts per sample to begin with, if I am interpreting Figure 1—figure supplement 1b correctly. I believe this means that DESeq2 is unlikely to be able to assign statistical significance for lower expression for these genes and that if it did, it would be prone to underestimating the fold-change if there are observations of zero counts. This "missing" or "misplaced" data in the lower left quadrant of the graphs could contribute to a (somewhat spurious) negative relationship and overemphasize this result.

(Related: Figure 1—figure supplement 3b has two p-values shown in red and black on each graph. I assume those are for two mutually exclusive subsets of the data. It should be explained in the legend.)

Second, the explanations and discussion given here are somewhat plausible and interesting, but they also seem incomplete and insufficient. For example, "This negative relationship is likely a by-product of increased mRNA abundances." Why is this? – I genuinely don't understand. As another example, "biophysical constraints" is a rather vague term. Are we talking about physical space inside cells? RNA polymerase abundance? DNA accessibility? I think at least an example or two of what the authors mean would need to be included. These additions would lead to much more discussion than this result probably warrants in terms of its overall importance to the manuscript (esp. if my statistical concerns are justified).

My recommendation is to remove this regression result from the paper.

*Reviewer #2 (Recommendations for the authors):*

The authors have adequately addressed the few comments I made in my initial review. I remain of the view that the results are somewhat oversold, especially with the several implied and direct references to the work establishing a mechanistic link to fitness (e.g., "To *understand* how different genomic changes lead to parallel fitness gains …" L74). I suggest the authors read through the work carefully to make sure that all promises are kept.

*Reviewer #3 (Recommendations for the authors):*

This study explores the role of parallel gene expression change (both transcriptional and translational) in the long-term evolution experiment (LTEE). The study represents an interesting look at parallel adaptation, helping contribute to our larger understanding of the biological level at which parallel changes take place and the predictability of adaptation to novel environments. I appreciate that the authors have explored parallelism from transcription, translation, and pathway levels down to nucleotide changes, and have included a phenotypic angle in the study.

I have read the manuscript first and then looked at the reviewer comments/responses to the reviewers. In my opinion, the authors have done a good job of addressing the previous reviewer's comments and the current version of this manuscript is well written and represents a significant contribution to the field. My only comment is that I wanted to see a 2D scatterplot of the lines in PC space in figure 1. Currently, Figure 1D summarises this analysis, but it could be made 50% smaller and a scatterplot of lines in PC space added (rather than putting it in the supplement).

---

## [Author Response]

[Editors’ note: the authors resubmitted a revised version of the paper for consideration. What follows is the authors’ response to the first round of review.]

Reviewer #1 (Recommendations for the authors):The authors present a detailed analysis of transcriptional (RNA-seq) and translational (Ribo-seq) changes occurring during the LTEE. The manuscript is generally well written and organized, and analyses seem to be appropriate. I have two general comments. First, it would help for the large amount of data to be presented with more focus to the motivating biological question. Second, which of the observed changes are cause, and which consequence, of changes in fitness.1. I found it hard to pin down the overarching goal of the work. A lot of data is presented, and it would be helpful for readers to understand the motivating questions that are being addressed. There are certainly many candidates. For example, 'bridging the gap between disparate genomic changes and parallel fitness gains' (L14), '[characterization of] the mechanistic basis of adaptation' (L39), '[exploration of] the role that transcription and translation play in increasing growth rates' (45). To me, some points seem difficult to pin down (what does 'bridging the gap' really entail?) while others oversell actual results (relatively little mechanism is presented, and, as far as I can see, nothing that directly connects any observed molecular phenotype change to a change in fitness/growth rate). I think the Introduction, and the manuscript generally, would be improved by identifying and presenting clear goals of the work, ideally related to significant biological questions that the work allows to be addressed. It is telling that the Discussion is very short, presenting almost no relationship between results presented here and previous work.

We thank the reviewer for their critical reading of our work and their constructive suggestions. We have rewritten and rearranged significant portions of the paper in order to present a clearer idea of the main goals of the work, as well as better partition the large amounts of data presented. We now have a section towards the end of the introduction detailing the specific questions we plan to address in the manuscript. We have also reworded ambiguous phrases such as “bridging the gap” to be more specific so the intent of the sentence is clear. Finally, we have also significantly expanded the Discussion section and added additional points in the Results section that contextualize our findings with other published work on the LTEE. We believe these changes have significantly improved the manuscript.

2. Many bacterial genes have growth rate-dependent expression – due, for example, to effects of cell growth on regulator concentrations (e.g., Cell 2009 139:1366). Such dependence will tend to create parallel gene expression changes in faster growing strains relative to a slower growing reference. Clearly, this effect doesn't explain all the changes observed in this study, but it is not clear how many it does explain, and, perhaps more importantly, how to interpret the possibility that a fraction of all expression changes are an effect rather than a cause of fitness increases. At minimum, I'd like the authors to present and discuss this point generally and, where appropriate, to discuss how it effects the specific conclusions they make (and the kind of questions that can reasonably be asked – e.g., growth rate dependent expression changes make it much more difficult to work back from expression changes to causal mutations).

We thank the reviewer for this comment and agree that this is an extremely important point that we had initially ignored. We have added a section in the Discussion that addresses this point. In short, we acknowledge the fact that some proportion of the differentially expressed genes observed in our datasets are going to be consequential to the fact that lines in LTEE are growing faster.

Reviewer #2 (Recommendations for the authors):This paper describes a systems biology study of how RNA expression levels and mRNA translation rates (ribosomal occupancy) changed as eleven *E. coli* populations adapted to laboratory conditions over 50,000 generations during the Lenski long-term evolution experiment (LTEE). The main finding is that the molecular phenotype-that is, which genes had altered expression and by how much-of the eleven independently evolved bacterial strains was more conserved than the underlying genetic changes-that is, which genes directly sustained mutations in the lineage leading to that strain during adaptation. Nearly all changes were at the level of transcriptional control of gene expression, with very few changes in how mRNAs were translated. These findings should be of broad interest because they inform theories about how gene expression evolves during adaptation.

We thank the reviewer for their overall positive impression of our manuscript and it’s appeal to a broad audience.

StrengthsThis study combines RNA-Seq and Ribo-Seq to look at changes in RNA abundance and mRNA translation rates. It represents a substantial advance, both in the techniques used and the scope in terms number of *E. coli* strains analyzed, relative to prior work that had analyzed two strains at 20,000 generations using an array-based technology close to twenty years ago. Other positive aspects of the study are that it uses spike-in controls for determining absolute RNA abundance and makes an effort to account for changes in the sizes of evolved cells. Changes in gene regulation are examined at the level of cellular processes but also profiled for specific regulons (e.g., malT and nadR). The finding of an apparent increase in the rate of translation termination is novel. Overall, the methods display a high degree of rigor. Finally, the citation and description of past results from the LTEE is thorough and appropriate. The authors should be commended for this, since this is a rather large amount of history to take into consideration, and they do not appear to have worked on this system before.

We are grateful to the reviewer for their acknowledgement of the tremendous amount of effort that has gone into this manuscript, the novelty of our findings, and the attention to rigor and details employed in reaching the final conclusions.

WeaknessesThere are some potential limitations/caveats related to the exact growth conditions used by the authors, which have a much higher concentration of the limiting nutrient (glucose) than was present during evolution of these strains in the LTEE. Some interpretation and analysis of how gene expression changes are related to fitness evolution could be improved. The correlation between large deletions and reduced gene expression of the genes contained within them could be examined in a different way that might lead to a significant effect. Inferences about the importance of how increases versus decreases in gene expression contribute to fitness evolution are indirect and do not appear to be completely justified.

We thank the reviewer for a critical reading of the manuscript and their feedback. We hope a significant rewriting and rearrangement of the manuscript has addressed many of their concerns. We address the specific concerns raised by the reviewer below.

1) Given that this was a major result, we felt that the paper could be improved by including a simple figure illustrating the similarity of the evolved strains to one another compared to how different they all are from the ancestor. We believe it would be easy to show this using a PCA plot. It is also possible that this analysis would show that some of the evolved lines are more like one another than others are, which may be interesting in light of some of our other recommendations.

We agree with the reviewer that figures showing the relationship between evolved lines would be a valuable addition manuscript. We performed PCA based on fold-changes in mRNA levels across all evolved lines and describe the findings in Figure S2 and accompanying text in the results. Briefly, the PCA reveals that evolved lines Ara-2 and Ara-3 show the largest distance in principal components relative to other lines. This is, perhaps, unsurprising given that Ara-3 has evolved the ability to aerobically metabolize citrate as a carbon source (Blount et al., 2012), and Ara-2 has developed distinct, coexisting ecotypes (Rozen et al., 2009). However, despite some changes in expression levels unique to these two lineages, the high overall correlation in fold-changes in mRNA expression levels suggest that most changes are shared and have occurred in parallel across evolved lineages (Figure 1D).

2) The manuscript largely assumes that fitness is the same across the eleven LTEE populations. While it is true that the fitness improvements in each population are very similar, both the Wiser et al. 2013 Science paper cited in the Introduction and the follow-up Lenski et al. 2015 Proc Royal Soc B paper, do show that there are also systematic differences between some populations. For example, the hypermutators do tend to have higher fitness than the others, on average. There is also the case of population Ara+1, which appears to be lagging in fitness because it sustained an unusually large number of transposon-mediated mutations (Consuegra et al. 2021 Nat Comm). Eleven clones are probably not enough to try to start predicting fitness from gene expression profiles, but it would be interesting if any global analysis of the data found that outliers in terms of gene expression were also outliers in terms of fitness.

We completely agree with the reviewer that it would be interesting to identify expression drivers of fitness differences between individual lines. While a complete analysis and prediction of fitness effects from gene expression profiles will require a detailed whole-cell mechanistic model (Macklin et al., 2020), PCA of fold-changes in expression levels can provide crude insights into whether there exist a set of genes whose expression changes while drive differences in growth rates. We find that while the PCA does separate some of the mutators from the non-mutators, the first two principal components are largely driven by expression profiles of Ara-3 and Ara-2, likely due to their unique phenotypes of citrate metabolism and distinct ecotypes, respectively. We’ve added this to the text in the Results subsection “Variation in expression changes across evolved lines”.

3) We think it is important to change the plotted per CFU values in Figure 1 to be per typical single cell volume. The filamentation observed in several of the evolved lineages dramatically affects the estimates of RNA abundance per CFU. While this is factually correct according to the methods and noted in a supplementary analysis paragraph, it would be much better to correct Figure 1 to use a different basis than CFUs, so that it does not give the impression that the per cell mRNA levels changed by >10-fold for some evolved strains. We believe the authors could correct this to be "per evolved cell volume", by estimating how many typical cell equivalents there are on average per CFU (filamented or not).

We thank the reviewer for this note about CFU and our metrics on estimating RNA abundance per CFU. While we agree with the reviewer that using CFU as a basis might give an incorrect impression on “per cell” estimates of RNA, using “per evolved cell volume” introduces additional challenges. Firstly, some colonies may have been formed by multiple cells and the number of cells per filament vary widely across lines. As a result, taking a median or mean cell-volume for each line can be misleading. This is evidenced by the fact that the median cell volume changes by ~3 fold but RNAs/CFU has a ~100 fold range. Therefore, dividing RNAs/typical-cell-volume will indicate that the density of RNAs per volume has increased dramatically. While we do expect an increase in density of RNAs, as not all macromolecules scale with cell-size, this apparent increase might be artificial due to our CFU measurements.

4) The observation that downregulated versus upregulated genes are more likely to show the same change in other lines is interesting (Line 208-212). However, we don't understand why these results indicate that there are "fewer genes and pathways whose downregulation increases fitness" necessarily. What is the connection to fitness?

We apologize for the poor phrasing of the sentence that led to confusion. The key observation here is that pathways that are downregulated are shared across lines more often than pathways that are upregulated. Assuming that changes in genes/pathways are adaptive, there might be a smaller set of genes/pathways whose downregulation consistently increases fitness thereby leading to higher observed parallelism in these pathways. On the other hand, since pathways that are upregulated tend to be somewhat unique to each line, indicating that there might be more diverse pathways whose upregulation might be adaptive. Due to the speculative nature of this analysis, we have removed it from the revised manuscript..

5) The test of the hypothesis that genes that were deleted in some lineages would be downregulated in other lineages in which they were not deleted gave a negative result. As the authors suggest, it may be that downregulation of just one of the deleted genes yields the fitness benefit for the entire deletion. In addition to the current analysis, we would recommend repeating the analysis in a way that tests this refined hypothesis that at least one deleted gene is downregulated in the other lines. It may be possible to identify which gene or genes "drove" the deletion and which genes were collateral deletions.

We thank the reviewer for suggesting a test to identify deletions that might be the driver of large deletions observed in LTEE. However, based on comments from other reviewers, and in an effort to present a coherent narrative in this work, we’ve chosen to remove this section of the paper. We have, however, worked hard to share the data and code in a machine-readable format with detailed documentation with the hope that other groups will be able to utilize them to test this and other interesting hypotheses.

6) *E. coli* cells used for RNA-Seq and Ribo-Seq were cultured in a slightly different medium than was used for the evolution experiment. The base media is the same, but a much higher concentrations of glucose was used (4,000 µg/mL versus 25 µg/mL). Presumably, this was necessary in order to be able to harvest enough cells for the RNAseq and ribosomal profiling experiment. Still, this difference should be noted and any affect that it might have on interpreting the data should be discussed. There also appears to be another minor difference in that the Lenski website recipe calls for thiamine supplementation.

We completely agree with the reviewer that a higher concentration of glucose used here could in principle introduce artifacts that might be hard to identify. The reviewer is correct in pointing out that this higher concentration was necessary to obtain enough cells for ribo-seq experiments. We have now added a section in the supplement “The potential effects of increased sugar in the culture medium” to highlight potential caveats. With regards to thiamine supplementation, all our media were supplemented with thiamine. This detail was inadvertently left out in the methods section and has been corrected in the revised version where we explicitly state that the bacteria were cultured in medium as per the recipe on the LTEE website (with the added note about glucose supplementation).

7) There are some deep genetic divergences and large phenotypic changes in some of the LTEE populations that make it important to know which type of clonal isolate was analyzed here to interpret the results. Most importantly, population Ara-3 evolved citrate utilization, which enables it to grow to a higher cell density. Is the clone that was analyzed Cit+ or Cit-? Also, population Ara-2 diverged into "large" and "small" colony types. What is the type of the clone that was analyzed from each of these populations? If a Cit+ clone was used, were cells harvested at an early enough point that gene expression reflects growth of these cells on glucose (or a glucose/citrate mixture) rather than solely on citrate?

We have added the details of the strains used in this study in the revised text. Briefly, We have a Cit+ Ara-3 and based on comparing mutations in our clone to those described in Plucain et al. 2014, our Ara-2 is of the L ecotype.

8) The Methods section should be revised. Currently, the quality and level of detail is very uneven. There are placeholders and mixed citation styles that make it look like some of this section was still in rough draft form. Certain sections may give too much information. For example, the RNAseq library preparation methods seem to be exactly from a standard NEBnext kit? It may be better to state the differences from the standard protocol, if any, in this section. Other sections seem to leave out important information. The cell size section discusses measuring the length of the cells, but the Results on line 98 focus on cell volume, not length. As another example, perhaps too much detail is given in terms of the gel run time on Line 537, but the key detail of what the fragment size range of the "region corresponding to the expected product size" that was excised from this gel is not provided on Line 538.

We thank the reviewer for pointing out the unevenness of details in our methods section. We have revised the methods section to reduce unnecessary details regarding library prep methods and expanded sections, such as the expected size of our library products, where key details were missing.

9) Several of the earlier Results subsections mention that a result from the RNA-Seq data was similar for the Ribo-Seq data before the Ribo-Seq data is fully described. It may be best to wait on making those comparisons and consolidate all of those statements under the Ribo-Seq section.

We agree with the reviewer and have reorganized both the text and figures accordingly.

10) Line 131: Define TPM in the main text here. Currently, it is only defined in the figure legend.

TPM is now defined in the text at its first mention.

11) Line 140: This statement appears to have a typo that results in an incorrect meaning: "each of the lines was founded on a unique set of mutations". Perhaps they mean that each of the lines accumulated a unique set of mutations during the LTEE?

We apologize for the confusing phrase, and have altered the statement to be clearer.

12) Line 180: This result is in Figure 2C instead of 2B.

This has been corrected.

13) Line 610: This should read "two samples of each evolved clone" rather than "2 evolved clones".

This has been corrected.

Reviewer #3 (Recommendations for the authors):The LTEE holds a special place in the history of evolutionary biology and there is value in learning more about this classic case study. As an evolutionary biologist (not a microbiologist), my primary interest is what does this study of expression add to our understanding of evolution within the LTEE. This work is descriptive rather than testing well-motivated hypotheses but the authors have unearthed some intriguing patterns.

We thank the reviewer for their detailed comments and suggestions on the manuscript. We agree with the reviewer that one of the primary goals of our work was to provide a valuable resource to enable testing of specific hypotheses in evolutionary biology. We have uncovered interesting patterns in these high-throughput datasets and tested several specific hypotheses. However, we agree that the writing, and the presentation of the hypotheses/data did not always make it obvious what was being tested. We have revised the manuscript from the ground-up and hope this revised version helps alleviate concerns that the reviewer brought up here. We address specific comments below.

1) Much of the emphasis in this work is on the extent to which changes in expression are parallel. I have several concerns on this front.a) What is the expectation about the degree of parallelism? In several places the authors refer to there being a "high" degree of parallelism. "High" compared to what? Zero? Is their null expectation for populations evolving in identical environments to have zero parallel expression changes? In Figure 3, they make a comparison of shared expression changes to shared genetic changes. Of course, a single genetic change (for example, in a transcription factor) could cause expression changes in many downstream genes so should not a higher number of shared expression changes than genetic changes be expected? I find it bothersome to be told that there is a "high" degree of parallelism when there is no expectation. Was it possible to observe 5 times more parallelism than they observed? Compared to that possibility, they observed a "low" degree of parallelism.

We thank the reviewer for this critique and completely agree with their points regarding the need for an appropriate null model to compare against the observed degree of parallelism in gene expression. In the revised manuscript we have added detailed analyses based on both standard statistical models and simulation results to address this issue. We first show that the null distribution of shared differentially expressed genes across evolved lines is well approximated by the Sum of Independent Non-Identical Binomial (SINIB) random variables (Liu and Quertermous, 2018) We show that the observed degree of parallelism in expression changes is truly remarkable. For instance, if differentially expressed genes were randomly distributed across all lines, we would expect ~3 altered genes to be shared across five or more lines. Instead we find that 117 genes have significant and consistent altered expression levels in at least five evolved lines.

b) A practical issue with parallelism is statistical power. Though statistical details are annoyingly scant throughout, it appears the authors have typically required a gene to exhibit "significant" evolutionary change in two (or more) lines to be considered parallel. Because statistical power to detect change in any one line is less than 100% (and for many changes will probably be more like 10%), the power to observe parallelism will be limited, perhaps severely, in many cases. Unfortunately, I have no sense of the extent to which parallelism in this study is underestimated because of this problem.

We are not sure what the reviewer means by the “statistical power to detect change in any one line is less than 100%” The statistical power in classifying any single gene to be differentially expressed is a direct function of read coverage for that gene. There are two points we would like to highlight here. First, the read coverage per gene in our sequencing data is quite high even after removing PCR duplicates (Figure S1). This is one of the reasons for the high correlation between biological replicates. Second, we designate a gene as differentially expressed at a stringent False Discovery Rate (FDR) cutoff (q-value < 0.01). Despite these stringent cutoffs, we find that about 270 genes are differentially expressed on average across all evolved lines. We have highlighted these details in the revised manuscript, and hope that this alleviates some of the concerns that the reviewer had regarding statistical power in our analysis.

With regards to when a gene should be considered to be parallely changed across lines, any threshold on the number of lines is going to be arbitrary. In the revised manuscript, we have refrained from calling anything as parallely altered unless it was observed in at least 4 lines.

So what are we left with? In many cases throughout the manuscript we can say there is some parallelism. I do not find the observation of some degree of parallelism particularly interesting or surprising, especially given what we already know about the LTEE. To me, the question is whether the degree of parallelism meets expectation or is remarkable and that question is not addressed.

We are sympathetic to the reviewer’s sentiments and hope that the revised manuscript has sufficiently addressed these concerns, and convinced the reviewer that the degree of parallelism in gene expression changes is quite remarkable (Figures 1E, and S2C).

2) The negative correlation between evolved change and ancestral expression level.This pattern shown in Figure 2C and D and Figure S6 is surprising to me. To me, this was one of the most important results and should have been more prominently featured. My a priori expectation would be no correlation, yet they observe this negative correlation in every line (Figure S6). The authors hypothesize that this may be because of a biophysical constraint ("maximally expressed" genes can only evolve lower expression and "minimally expressed" genes can only evolve increased expression). That hypothesis should certainly be considered. I have no better hypothesis but I find it surprising if there are enough genes close to these limits to drive this pattern. From Figure S6 it does not appear this pattern is driven by genes at the range limits of ancestral expression. Is there work in *E. coli* about whether a reasonable fraction of genes are at an expression limit? (For genes at the high expression limit, presumably gene duplication can occur as a means to increase expression. I realize such mutations would be rare relative to point mutations and small indels.) The reported pattern would be even more compelling if they had some additional means to evaluate the biophysical constraint hypothesis.

We completely agree with the reviewer that this is indeed a very interesting pattern, and one that we hadn’t anticipated prior to this analysis. In the revised manuscript, we have significantly expanded the discussion on biophysical constraints on gene expression in the Results subsection “Magnitude and direction of expression changes”.

3) Parallelism of up- vs. down-regulated expressionThough there are problems with expectations for the extent of "parallelism", this issue is avoided in some cases. One intriguing case is for parallelism in genes that evolve up- vs. down-regulated expression. The authors report that "more downregulations were shared across lines than upregulations". To me, this was one of the most interesting patterns they reported because there is a simple expectation of equality between up and down regulation. However, I have several concerns.a) It wasn't clear to me there was any statistical test of this claim.

In the revised manuscript, we have added the following text along with the appropriate statistical tests for these claims.

“While the number of DEGs vary widely across lines (Figure S2B), we find that in 7 out of 11 evolved lines, significantly more genes were downregulated than upregulated (Binomial test, p-value < 0.05). Furthermore, the magnitude of fold-changes of downregulated DEGs were significantly higher than fold-changes of upregulated DEGs in all 11 evolved lineages (KS-test, p-value < 0.0001) (Figure S2D).”

b) It seems that one should control for the number of down vs- up regulated genes (i.e., if there are 500 down regulated genes and 200 up-regulated ones in Line 1, then it would not be particularly interesting we found that Line 2 shared 50 of the down-regulated ones but only 20 of the up-regulated ones

We thank the reviewer for bringing this important point to our attention. In the revised statistical tests for parallelism in expression changes, we use the SINIB model that explicitly takes into account the differences in the number of up/down-regulated genes in each line (Figure 1D, S2C).

We hope that this alleviates the reviewer’s concern.

c) The authors also report that down-regulated changes tend to larger in magnitude than up-regulated ones. This is a key piece of information. With respect to the power comment above (1b) this means that there will be greater power to detect parallel changes for up-regulated vs. down-regulated changes.I suggest using a statistical model that attempts to account for these issues.

We agree with the reviewer that given the smaller magnitude of up-regulated genes, we might be able to identify a smaller set of significantly altered genes at a particular statistical threshold (q-value < 0.01). To address this issue, we quantify the degree of parallelism using the SINIB model in both down- and up-regulated genes separately. Despite the differences in effect sizes and number of differentially expressed genes, for both these groups of genes, we find a significantly higher number of shared DEGs than expected by chance (Figure 1E and S2C).

4) Importance of transcription vs translation to expression evolutiona) The authors find highly correlated values between RNAseq and Riboseq data. This is not at all surprising. Such correlations at the genome level are not useful because the among-gene variation in expression is so large (I have a similar complaint with lines 150-153).

We agree and following the reviewer’s advice, we have removed correlations between RNA-seq and Riboseq expression levels (transcript-per-million (TPM) correlations). We now report correlations in fold-changes in these two datasets instead, which removes the effects of among-gene variation in expression levels.

b) The authors use Riborex to test for evolved differences in ribosomal densities and find very few changes. They use this as a basis for arguing that transcription is far more important than translation for expression evolution. They may well be correct. However, I suspect that the statistical power to detect changes in ribosomal density is much lower than for detecting transcription changes (e.g., ribosomal density changes are hampered by measurement error in BOTH RNAseq and RIBOseq data). Given that the power to detect ribosomal density changes is almost certainly much lower than for transcription changes, it seems premature to make much of a claim about transcription vs. translation.

We agree, in principle, that the statistical power in detecting changes in ribosomal densities is lower than detecting changes at either scales independently. However, in our experience of working with riboseq data across a wide-range of species with similar sequencing depth (and in some cases lower depths), we still find hundreds of genes with altered ribosome-densities. In that sense, the very small number of altered genes here is surprising. Nonetheless, we have rewritten this section to moderate this claim.

5) Faster translation terminationFigure 4C is quite dramatic. That is a very interesting result.a) This another example where too few information is provided. What is the statistical test behind the p-value shown? Based on the figure, there is a highly significant difference but I suspect the test they did was wrong. Did they run a model that accounts for both "line" and the same codon represented in every line? I suspect they did a two-sample t-test and thus have an inflated degrees of freedom.

We thank the reviewer for pointing out this incorrect statistical test. Originally, we did employ a two-sample t-test, which does not account for the non-independence between lines. We have updated this analysis to compare changes (i.e. evolved vs. ancestral lines) between the ribosome-densities of sense-codons and stop-codons using a linear mixed model with random (i.e. line-specific) effects. We find that these new results are consistent with our original claim that stop-codons have significantly lower ribosome-densities compared to the sense codon ribosome-densities Nonetheless, the magnitude of this effect is variable across lines (decreases ranged from -0.088 to -0.657 log fold change in ribosome densities relative to the sense codons). This indicates that all evolved lines experienced an increased translation termination rate (relative to the ancestral line), but some evolved lines experienced a greater increase in termination rate than others.

b) Naively, I would have thought there would always be selection favouring faster translation termination (not just in the LTEE). Is there some plausible reason for why there should selection for faster translation termination in the LTEE than in the many millions of years prior to the start of the experiment?

While faster translation termination may increase ribosome recycling and enable faster growth, it comes at the expense of a loss of a key regulatory mechanism in translational control. As a result, it remains unclear if these regulatory changes can evolve in more complex environments.

With respect to my comment 3, here is a potential way to improve these issues: Using all genes that have evolved significantly in at least 1 line, run something like the following statistical model:model <- glm(cbind(Nsig, 11 – Nsig) ~ UpOrDown + Magnitude + AncExpression, family = binomial)where Nsig is the number of lines where it significantUpOrDown is an indicator whether expression evolved up or downMagnitude is the |Log2FC| averaged over only the lines where it evolved significantly (because those are the lines that determined the genes inclusion in the set for this analysis)AncExpression is the Log(TPM) in the ancestor.The last two terms are an attempt to control for power to detect parallelism. It won't be perfect but it will be a considerable improvement over the current version.

We thank the reviewer for suggesting a possible solution to test for significant parallelism. We have employed a different approach as outlined earlier (SINIB) that we hope will prove sufficient.

Figure 2C. "…only statistically significant" genes. Significant in what comparison? In any 1 line vs. ancestor?

We have altered the text to make it clear what we mean when we say statistically significant.

References

Blount ZD, Barrick JE, Davidson CJ, Lenski RE. 2012. Genomic analysis of a key innovation in an experimental *Escherichia coli* population. *Nature* 489:513–518. doi:10.1038/nature11514

Liu B, Quertermous T. 2018. Approximating the sum of independent non-identical binomial random variables. *R J* 10:472. doi:10.32614/rj-2018-011

Macklin DN, Ahn-Horst TA, Choi H, Ruggero NA, Carrera J, Mason JC, Sun G, Agmon E, DeFelice MM, Maayan I, Lane K, Spangler RK, Gillies TE, Paull ML, Akhter S, Bray SR, Weaver DS, Keseler IM, Karp PD, Morrison JH, Covert MW. 2020. Simultaneous cross-evaluation of heterogeneous *E. coli* datasets via mechanistic simulation. *Science* 369. doi:10.1126/science.aav3751

Rozen DE, Philippe N, Arjan de Visser J, Lenski RE, Schneider D. 2009. Death and cannibalism in a seasonal environment facilitate bacterial coexistence. *Ecol Lett* 12:34–44. doi:10.1111/j.1461-0248.2008.01257.x

[Editors’ note: further revisions were suggested prior to acceptance, as described below.]

The manuscript has been improved but there are some remaining issues that need to be addressed, as outlined below:As you will see from the comments below, the reviewers were largely happy with the significant revisions you have provided. I agree with reviewer 1 that the regressions are problematic, as you are underpowered for genes with low expression. Please do add the PCA, as suggested by Reviewer 3.

We thank the editor for their supportive comments and suggestions. We have made all the edits suggested by the reviewers and look forward to the manuscript’s acceptance for publication.

Reviewer #1 (Recommendations for the authors):The current draft of this article is improved, though it still feels a little disjointed and meandering. It lacks the razor focus that takes one from a research question in the introduction, to the most relevant results addressing that question, and finally to a discussion of those results in the context of the field and other studies. That said, these are very interesting and complex datasets. The article communicates a number of interesting findings clearly, and it makes the datasets available so that others can continue to analyze them.1) I found the paragraph beginning on Line 146 describing the negative relationship between fold-change in evolved gene expression and gene expression level in the ancestor to be problematic in two ways.First, I worry about how possible limitations in what types of expression changes can be detected may affect the regressions in Figure 1—figure supplement 3b. For highly expressed genes, I don't doubt that there is more downregulation than upregulation. But, for lowly expressed genes, a smaller initial number of counts will limit how large of a negative fold-change could be observed at some point and whether such a change can be judged as statistically significant (and therefore included in the regression). It seems like a non-negligible number of genes only have <10 counts per sample to begin with, if I am interpreting Figure 1—figure supplement 1b correctly. I believe this means that DESeq2 is unlikely to be able to assign statistical significance for lower expression for these genes and that if it did, it would be prone to underestimating the fold-change if there are observations of zero counts. This "missing" or "misplaced" data in the lower left quadrant of the graphs could contribute to a (somewhat spurious) negative relationship and overemphasize this result.(Related: Figure 1—figure supplement 3b has two p-values shown in red and black on each graph. I assume those are for two mutually exclusive subsets of the data. It should be explained in the legend.)Second, the explanations and discussion given here are somewhat plausible and interesting, but they also seem incomplete and insufficient. For example, "This negative relationship is likely a by-product of increased mRNA abundances." Why is this? – I genuinely don't understand. As another example, "biophysical constraints" is a rather vague term. Are we talking about physical space inside cells? RNA polymerase abundance? DNA accessibility? I think at least an example or two of what the authors mean would need to be included. These additions would lead to much more discussion than this result probably warrants in terms of its overall importance to the manuscript (esp. if my statistical concerns are justified).My recommendation is to remove this regression result from the paper.

We thank the reviewer for their detailed explanation regarding concerns about power to detect negative fold-changes for lowly expressed genes. We agree that low read counts of poorly-expressed genes hamper our ability to perform this analysis and this is at least partially responsible for the relationship we observe. Following the reviewer’s recommendation, we removed this analysis from the manuscript.

Reviewer #2 (Recommendations for the authors):The authors have adequately addressed the few comments I made in my initial review. I remain of the view that the results are somewhat oversold, especially with the several implied and direct references to the work establishing a mechanistic link to fitness (e.g., "To *understand* how different genomic changes lead to parallel fitness gains …" L74). I suggest the authors read through the work carefully to make sure that all promises are kept.

We thank the reviewer for bringing this to our attention. We have softened the language around some of our results and tried to clarify when we are speculating a relationship versus claiming the existence of one.

Reviewer #3 (Recommendations for the authors):This study explores the role of parallel gene expression change (both transcriptional and translational) in the long-term evolution experiment (LTEE). The study represents an interesting look at parallel adaptation, helping contribute to our larger understanding of the biological level at which parallel changes take place and the predictability of adaptation to novel environments. I appreciate that the authors have explored parallelism from transcription, translation, and pathway levels down to nucleotide changes, and have included a phenotypic angle in the study.I have read the manuscript first and then looked at the reviewer comments/responses to the reviewers. In my opinion, the authors have done a good job of addressing the previous reviewer's comments and the current version of this manuscript is well written and represents a significant contribution to the field. My only comment is that I wanted to see a 2D scatterplot of the lines in PC space in figure 1. Currently, Figure 1D summarises this analysis, but it could be made 50% smaller and a scatterplot of lines in PC space added (rather than putting it in the supplement).

We thank the reviewer for their supportive comments. Following reviewer’s suggestion, we have moved the PCA plot from the supplement to figure 1F.